



# Development of East Asia Regional
# Reanalysis based on advanced hybrid gain
# data assimilation method and evaluation
# with E3DVAR, ERA-5, and ERA-Interim
# reanalysis
Eun-Gyeong Yang, Hyun Mee Kim[*], and Dae-Hui Kim
*Atmospheric Predictability and Data Assimilation Laboratory*
*Department of Atmospheric Science, Yonsei University, Seoul, Republic of Korea*

[*] *Corresponding author address*: Hyun Mee Kim, Department of Atmospheric Sciences, Yonsei University, 50 Yonsei-ro, Seodaemun-gu, Seoul, 03722, Republic of Korea.

E-mail: khm@yonsei.ac.kr





**ABSTRACT**

The East Asia Regional Reanalysis (EARR) system is developed based on the advanced
hybrid gain data assimilation method (AdvHG) using Weather Research and Forecasting (WRF)
model and conventional observations. Based on EARR, the high-resolution regional reanalysis
and reforecast fields are produced with 12 km horizontal resolution over East Asia for 2010–
2019. The newly proposed AdvHG is based on the hybrid gain approach, weighting two
different analysis for an optimal analysis. The AdvHG is different from the hybrid gain in that
1) E3DVAR is used instead of EnKF, 2) 6 h forecast of ERA5 is used to be more consistent
with WRF, and 3) the pre-existing, state-of-the-art reanalysis is used. Thus, the AdvHG can be
regarded as an efficient approach to generate regional reanalysis dataset due to cost savings as
well as the use of the state-of-the-art reanalysis. The upper air variables of EARR are verified
with those of ERA5 for January and July 2017 and the two-year period of 2017-2018. For upper
air variables, ERA5 outperforms EARR over two years, whereas EARR outperforms (shows
comparable performance to) ERA-I and E3DVAR for January in 2017 (July in 2017). EARR
better represents precipitation than ERA5 for January and July in 2017. Therefore, though the
uncertainties of upper air variables of EARR need to be considered when analyzing them, the
precipitation of EARR is more accurate than that of ERA5 for both two seasons. The EARR
data presented here can be downloaded from https://doi.org/10.7910/DVN/7P8MZT for data
on pressure levels and https://doi.org/10.7910/DVN/Q07VRC for precipitation.

## 1. Introduction

Reanalysis datasets have been widely used in the socio-economical field as well as meteorological and climate research areas all over the world. Most of reanalysis datasets consist of global reanalysis whose spatial and temporal resolutions are relatively coarse (e.g., Schubert et al. 1993; Kalnay et al. 1996; Gibson et al. 1997; Kistler et al. 2001; Kanamitsu et al. 2002; Uppala et al. 2005; Onogi et al. 2007; Bosilovich 2008; Saha et al. 2010; Dee et al. 2011; Rienecker et al. 2011; Bosilovich 2015; Kobayashi et al. 2015; Hersbach et al. 2020). As the importance of regional reanalysis dataset emerged, many operational centers and research institutes around the world have been producing the dataset in their own areas (Mesinger et al. 2006; Renshaw et al. 2013; Borsche et al. 2015; Bromwich et al. 2016; Jermey and Renshaw 2016; Zhang et al. 2017; Bromwich et al. 2018; Fukui et al. 2018; Ashrit et al. 2020).

As part of this effort, regional reanalysis over East Asia were produced based on the Unified Model for the two-year period of 2013-14 and it was confirmed that regional reanalysis over East Asia is beneficial (Yang and Kim 2017; Yang and Kim 2019). However, because UM was no longer available for generating regional reanalysis over East Asia, another numerical weather prediction (NWP) model and its data assimilation (DA) method are required.

To find the most appropriate and cost-efficient DA method for a regional reanalysis over East Asia, several DA methods were compared. Yang and Kim (2021) demonstrated that the hybrid ensemble-variational data assimilation method (E3DVAR) shows the better performance compared to three-dimensional variational data assimilation (3DVAR) and ensemble Kalman filter (EnKF) over East Asia for January and July in 2016. However, it is essential to confirm if this hybrid method is accurate enough to be used for a regional reanalysis over East Asia. Thus, E3DVAR was compared with the latest and the previous reanalysis data from ECMWF (i.e., ECMWF's fifth-generation reanalysis (ERA5, Hersbach et al. 2020) and

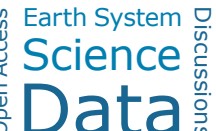

ERA-Interim (ERA-I, Dee et al. 2011)) for (re)analysis and (re)forecast variables and it was
found that a performance for a regional reanalysis needs to be further improved.
For this reason, a new advanced hybrid gain (AdvHG) data assimilation method, which
combines E3DVAR and ERA5 based on WRF model, is newly proposed and investigated in
this study. A hybrid gain data assimilation method has been developed as a new kind of hybrid
methods (Penny 2014). Based on this method, an advanced data assimilation method is newly
developed in this study. Finally, using this newly proposed DA method (AdvHG), East Asia
regional reanalysis (EARR) system is developed based on WRF model. EARR datasets have
been produced for ten-year period of 2010-2019 and are verified for two-year period of 2017–

2018.

To investigate the accuracy and uncertainty of the state-of-the-art AdvHG DA algorithm
developed in this study, analysis and forecast atmospheric variables of E3DVAR, AdvHG,
WRF-based ERA-I, and WRF-based ERA5 are evaluated for January and July in 2017,
respectively. In addition, reforecast precipitation fields of ERA-I and ERA5 from ECMWF are
also verified and compared. In section 2, the EARR system including model, data assimilation
method, and observations are explained. In section 3, the evaluation methods are presented.
The verification results of (re)analysis and (re)forecast variables are presented in section 4.
Section 4.1 presents evaluation results for wind, temperature, and humidity variables, and
section 4.2 presents those for precipitation (re)forecast. Section 5 presents data availability.
Lastly, summary and conclusions are presented in section 6.

## 81  2. Reanalysis system

*2.1. Model*
In this study, the Advanced Research Weather Research and Forecasting (WRF, v3.7.1)
model is used with 12-km horizontal resolution (540 x 432 grid points) and 50 vertical levels





(up to 5 hPa) as shown in Fig. 1. The model settings and physics scheme are summarized in
Table 1. Analysis fields are obtained every 6 h (00, 06, 12, and 18 UTC) via assimilation of
conventional observations with a 6 h assimilation window, and forecast fields are integrated up
to 36 h. The ERA5 reanalysis (Hersbach et al. 2020) is used as the first initial condition before
the cycling, and as boundary conditions every 6 h.
*2.2. Data assimilation methods*
*2.2.1. E3DVAR*
The E3DVAR method is one of hybrid data assimilation methods, which use a static
climatological background error covariance (BEC) and ensemble-based flow-dependent BEC,
and couples the EnKF and 3DVAR (Zhang et al. 2013). E3DVAR is based on a cost function
of 3DVAR. In E3DVAR, EnKF provides flow-dependent BEC as well as updates perturbations
for ensemble members. Following Zhang et al. (2013),

$$J^b = J_s^b + J_e^b = \frac{1}{2}\delta\mathbf{x}^{\mathrm{T}}\left[(1-\beta)\mathbf{B} + \beta\mathbf{P}^f \circ \mathbf{C}\right]^{-1}\delta\mathbf{x} \ , \qquad (1)$$

where $J_s^b$ is a traditional cost function based on a static climatological BEC $\mathbf{B}$ and $J_e^b$ is an
additional cost function based on ensemble-based BEC $\mathbf{P}^f$. $\mathbf{C}$ is a correlation matrix for
localization of the ensemble covariance $\mathbf{P}^f$. The weighting coefficient $\beta$ between static and
ensemble-based BEC is set to 0.8 in this study. To account for model error for E3DVAR, multi-
physics scheme is applied to 40-member ensembles. Yang and Kim (2021) found that E3DVAR
is the most appropriate DA method among 3DVAR, EnKF, and E3DVAR methods over East
Asia. More detailed information on E3DVAR implemented in this study can be found in Yang
and Kim (2021).
*2.2.2. Advanced hybrid gain data assimilation method*
In the last decade, the traditional hybrid methods have been widely used for many
operational centers and research institutes. Recently, Penny (2014) has proposed a new class



of hybrid gain methods combining desirable aspects of both variational and EnKF families of
algorithms by weighting analyses from 3DVAR and LETKF for an optimal analysis in the
Lorenz 40-component model. Since then, this algorithm has been implemented at ECMWF
(Bonavita et al. 2015) and at a hybrid global ocean DA system in National Centers for
Environmental Prediction (NCEP) (Penny et al. 2015).

The hybrid gain algorithm can be described with the following equations:

$$\mathbf{x}^a_{Hyb} = \alpha\mathbf{x}^a_{\mathrm{det}} + (1-\alpha)\overline{\mathbf{x}^a} \,, \qquad (2)$$

where $\mathbf{x}^a_{\mathrm{Hyb}}$, $\mathbf{x}^a_{\mathrm{det}}$, and $\overline{\mathbf{x}^a}$ denote the hybrid analysis, deterministic analysis, and the ensemble
mean analysis from the ensemble-based assimilation method, and $\alpha$ is a tunable parameter
(Penny 2014, Houtekamer and Zhang 2016).

The hybrid gain method is different from traditional hybrid methods, in that a hybrid gain

approach linearly combines analysis fields from EnKF and variational DA method to produce
a hybrid gain analysis rather than linearly combining respective BECs (Penny 2014). Basically,
the hybrid gain method is to hybridize two different Kalman gain matrices of ensemble-based
[Eq. (4)] and variational data assimilation system [Eq. (5)] as in Eq. (3).

$$\hat{\mathbf{K}} = \beta_1\mathbf{K}^f + \beta_2\mathbf{K}^B + \beta_3\mathbf{K}^B\mathbf{H}\mathbf{K}^f \,, \qquad (3)$$

where

$$\mathbf{K}^f = \mathbf{P}^f\mathbf{H}^\mathrm{T}(\mathbf{H}\mathbf{P}^f\mathbf{H}^\mathrm{T} + \mathbf{R})^{-1}, \qquad (4)$$

$$\mathbf{K}^B = \mathbf{B}\mathbf{H}^\mathrm{T}(\mathbf{H}\mathbf{B}\mathbf{H}^\mathrm{T} + \mathbf{R})^{-1}. \qquad (5)$$

By choosing the specific coefficients ($\beta_1$=1, $\beta_2 = \alpha$, $\beta_3 = -\alpha$), it can be written as in Eq. (6)
and it can give an algebraically equivalent result with Eq. (2) (Penny 2014).

$$\hat{\mathbf{K}} = \mathbf{K}^f + \alpha\mathbf{K}^B(\mathbf{I} - \mathbf{H}\mathbf{K}^f) \,. \qquad (6)$$



One of advantages of the hybrid gain algorithm with respect to its development is that pre-
existing operational systems can be used without significant modification for a hybrid analysis
(Penny 2014) and independent parallel development of respective methods is allowed
(Houtekamer and Zhang 2016). Furthermore, the hybrid gain approach can be considered as a
practical and straightforward method in the foreseeable future to combine advantageous
features of both ensemble- and variational-based DA algorithms (Houtekamer and Zhang 2016).
More detailed information on this algorithm can be found in Penny (2014).
In this study, based on the hybrid gain approach, an advanced hybrid gain data assimilation
method (AdvHG) is newly proposed as follows:

$$\mathrm{X}_{\mathrm{AdvHG}}^{a} = \alpha \mathrm{X}_{\mathrm{ERA5}}^{f(6h)} + (1-\alpha)\overline{\mathrm{X}}_{\mathrm{E3DVAR}}^{a}, \qquad (7)$$

where $\mathrm{X}_{\mathrm{ERA5}}^{f(6h)}$ denotes the 6 h forecast of ERA5 reanalysis based on WRF model and $\overline{\mathrm{X}}_{\mathrm{E3DVAR}}^{a}$
denotes the analysis of E3DVAR. This advanced hybrid gain approach is different from the
hybrid gain approach in that 1) E3DVAR analysis is used instead of EnKF, 2) 6 h forecast of
ERA5 is used instead of deterministic analysis from variational DA method, and 3) the pre-
existing and state-of-the-art reanalysis data (i.e., ERA5) is simply used instead of producing
deterministic analysis by assimilation. The reasons for these different approaches proposed in
this study are as follows:
1) E3DVAR is used instead of EnKF because Yang and Kim (2021) confirmed that
E3DVAR outperforms EnKF for winter and summer seasons over East Asia.
2) Instead of deterministic analysis, the 6 h forecast of ERA5 based on WRF model is
used to make the hybrid analysis more balanced and consistent with WRF model, because
ERA5 reanalysis fields are based on its own modeling system with coarser resolution, which
is different from that of this study.
3) European Centre for Medium-Range Weather Forecasts (ECMWF) reanalysis (ERA5)



is used instead of producing our own analysis fields from a variational DA method. This is a
very efficient approach because of the cost savings as well as the use of the high-quality latest
reanalysis from ECMWF assimilating all currently available observations with the state-of-the-
art and advanced technology.

Therefore, the approach proposed in this study is called as "advanced hybrid gain method"

(denoted as "AdvHG").
*2.3. Observations*

The NCEP PrepBUFR conventional observations (global upper air and surface weather

observations, NCEP/NWS/NOAA/U.S.DOC 2008) are used every 6 h (00, 06, 12, and 18 UTC)
for an assimilation by E3DVAR and AdvHG methods. The assimilated observations are as
follows: the surface observations (SYNOP, METAR, Ship, and Buoy), radiosonde observation
(SOUND), upper-wind report (PILOT), wind profiler, aircraft, atmospheric motion vector
(AMV) wind from a geostationary satellite (GEOAMV), and quick scatterometer (QuikSCAT).
All observations are spatially thinned by 20 km except for AMV thinned by 200 km as done
by Warrick (2015), Cotton et al. (2016), and Shin (2016).

To evaluate 6 h accumulated precipitation simulated by E3DVAR, AdvHG, ERA-I, and

ERA5 over East Asia, global surface weather observations (NCEP PrepBUFR,
NCEP/NWS/NOAA/U.S.DOC 2008) are used every 6 h (00, 06, 12, and 18 UTC). For an
evaluation of the monthly precipitation fields, the world monthly surface station climatology
(NCDC/NESDIS/NOAA/U.S.DOC et al. 1981) over 4700 different stations (2600 in more
recent years) is used.
*2.4.Global reanalysis datasets*

To compare EARR generated with other reanalysis datasets, ERA5 (Hersbach et al. 2020)

and ERA-I (Dee et al. 2011) reanalysis are chosen. The horizontal resolutions of ERA-I and
ERA5 are approximately 79 km and 31 km, respectively. Because ERA5 is based on the





operational system in 2016, improvements in model physics, numerics, data assimilation, and
additional observations over the last decade are the advantages of ERA5 (Hersbach et al. 2018).
Because reforecast as well as reanalysis fields are verified in this study, for forecast fields,
two different forecast fields from ECMWF (i.e., forecast based on WRF model and reforecast
based on ECMWF model) are used. The WRF forecast fields (i.e., WRF-based ERA5, WRF-
based ERA-I) using ERA5 and ERA-I as initial conditions are integrated with 12 km resolution.
Secondly, reforecast fields based on ECMWF model (i.e., ERA5_fromECMWF, ERA-
I_fromECMWF), provided and downloaded from ECMWF, are used.

## 181   3. Evaluation method

*3.1. Equitable threat score and Frequency bias index*
Based on the contingency table (Table 2), ETS is defined as

$$\text{ETS} = \frac{A - A_r}{A + B + C - A_r}, \text{ where } A_r = \frac{(A+B)(A+C)}{A+B+C+D}. \tag{8}$$

The ETS range is from -1/3 to 1 and the value 1 for ETS is a perfect score.
FBI is defined as

$$\text{FBI} = \text{Bias} = \frac{A+B}{A+C}. \tag{9}$$

The FBI indicates whether the model tends to over-forecast (too frequently, FBI>1) or under-
forecast (not frequent enough, FBI<1) events with respect to frequency of occurrence.
*3.2 Probability of detection and False alarm ratio*
Based on the contingency table (Table 2), POD is defined as

$$\text{POD} = \frac{A}{A+C} = \frac{\text{Hits}}{\text{Hits} + \text{Misses}}. \tag{10}$$

The POD range is from 0 to 1. POD is required to be used with FAR, because POD can be
artificially improved by systematically over-forecasting the events (Wilson 2010).



FAR is defined as

$$\text{FAR} = \frac{B}{A+B} = \frac{\text{False alarms}}{\text{Hits} + \text{False alarms}}. \qquad \textbf{(11)}$$

The range of FAR is from 0 to 1 and its lower score implies a higher accuracy.
*3.3 Brier skill score*
Verification of the performance of high-resolution forecast with the traditional verification
metrics (e.g., ETS, FBI) can be misleading due to double penalty, particularly for highly
variable fields (e.g., precipitation). Therefore, as one of spatial verification approaches that do
not require forecast to match point observation spatially, neighborhood (fuzzy) verification
method, which assumes that slightly displaced forecast can be acceptable and a local
neighborhood can define the degree of allowable displacement (Ebert 2008; Kim et al. 2015;
On et al. 2018), is used in this section. According to Ebert (2008), depending on the matching
strategy, neighborhood verifications can be categorized into two frameworks: 'single
observation-neighborhood forecast (SO-NF)' where neighborhood forecasts surrounding
observations are considered, and 'neighborhood observation-neighborhood forecast (NO-NF)'
strategies where not only neighborhood forecasts but also neighborhood observations
surrounding observations are considered. Due to the absence of high-resolution gridded
precipitation observation data in East Asia, various verification scores widely used as
'neighborhood observation-neighborhood forecast (NO-NF)' strategy are not available in this
study. Thus, in this section, Brier skill score as one of 'single observation-neighborhood
forecast (SO-NF)' strategy is introduced.
The Brier score (BS) is similar to the mean-squared error (MSE) and is defined as (Wilks

2006):

$$\text{BS} = \frac{1}{N}\sum_{i=1}^{N}(p_i - o_i)^2. \qquad \textbf{(12)}$$



where $p_i$ denotes the probability forecast, and $o_i$ denotes the binary observation which is either
0 or 1, and $N$ is the total number of observations during the given period. Generally, Brier skill
score (or Brier score) is used to verify ensemble forecasts which are able to calculate
probabilistic forecasts (Kay et al. 2013; Kim and Kim 2017). However, Brier skill score can
also be used for deterministic forecasts using a pragmatic post-processing procedure (Theis et
al., 2005; Mittermaier et al. 2014), which derives probabilistic forecasts from deterministic
forecasts at every model grid point by considering neighborhood forecast as *pseudo ensemble*.

$$\mathrm{BSS} = 1 - \frac{\mathrm{BS}}{\mathrm{BS_{ref}}}, \qquad (13)$$

where $\mathrm{BS_{ref}}$ is Brier score of reference. Brier skill score is skill score with respect to Brier score
as in Eq. (13). For reference, a climatology or other forecast can be used either. In this study,
the WRF-based ERA-I is considered as a reference.
*3.4 Pattern correlation coefficient*
The pattern correlation coefficient (PCC) is defined as Eq. (14) (Shiferaw et al. 2018; Yoo
and Cho 2018; Park and Kim 2020).

$$\mathrm{PCC} = \frac{\sum_{i=1}^{N}(x_i - \bar{x})(o_i - \bar{o})}{\left[\sum_{i=1}^{N}(x_i - \bar{x})^2 \sum_{i=1}^{N}(o_i - \bar{o})^2\right]^{1/2}}, \qquad (14)$$

where $x_i$ and $o_i$ are (re)forecast and observed precipitation at $i$th observation location and the
over-bar indicates the averaged variables over N observed stations in the verification area.

# 4. Results

*4.1 Evaluation of wind, temperature, and humidity variables*
*4.1.1   RMSE for January and July 2017*
The analysis and forecast RMSEs of E3DVAR, AdvHG, the WRF-based ERA-I, and



WRF-based ERA5 are calculated for zonal wind, meridional wind, temperature, and Qvapor
(water vapor mixing ratio in WRF) variables against sonde observations at 00 and 12 UTC for
January and July in 2017 and averaged over each month (Figs. 2, 3, and 4).

For analysis RMSE (Fig. 2), ERA5 is smaller than ERA-I for all levels and variables. In

particular, the analysis RMSE difference between ERA5 and ERA-I is distinctive for wind. The
vertically averaged wind RMSE of ERA5 for January (2.22 m s$^{-1}$) and July (1.98 m s$^{-1}$) in 2017
is smaller by approximately 0.23 and 0.3 m s$^{-1}$ than that of ERA-I for January (2.45 m s$^{-1}$) and
July (2.28 m s$^{-1}$) in 2017. The analysis RMSE of E3DVAR is smaller than that of AdvHG for
all pressure levels and variables, except for temperature in July at 1000 hPa and Qvapor in
January and July at 1000 hPa. In general, the analysis RMSE of AdvHG for all variables is
comparable to or greater than that of ERA5.

Regarding wind variables of analysis (Figs. 2a, b, c, and d), E3DVAR is the most closely

fitted to observations except for the wind in upper troposphere in January, followed by ERA5,
AdvHG, and ERA-I. For temperature RMSE (Figs. 2e and f), E3DVAR is smaller than AdvHG
and ERA5 is smaller than ERA-I. However, in January (Fig. 2e), ERA5 RMSE is the smallest
for upper troposphere and RMSEs of ERA5 and E3DVAR are similar to each other for lower
troposphere. In July (Fig. 2f), overall E3DVAR RMSE is the smallest except for 1000 hPa. For
Qvapor, RMSE in July is much larger than that in January due to a monsoonal flow carrying
moist air to East Asia. In general, Qvapor RMSE of E3DVAR is the smallest, followed by
ERA5, AdvHG, and ERA-I. Therefore, for all variables, generally E3DVAR analysis fields are
the most closely fitted to observations. Since the analysis RMSE implies how much analysis
fields are fitted to observations rather than the accuracy of analysis itself, not only analysis
RMSE but also forecast RMSE should be considered.

For 24 h forecast RMSEs (Fig. 3), ERA5 RMSE is the smallest for all levels and variables

for January and July in 2017. In January (Figs. 3a, c, e, and g), overall, the 24 h forecast RMSE





of ERA5 is the smallest and that of ERA-I is the largest for all variables, and RMSEs of AdvHG
and E3DVAR are greater than those of ERA5 and smaller than those of ERA-I. Regarding
AdvHG and E3DVAR, in general, AdvHG is smaller than E3DVAR for all levels and variables.
Thus, in January, ERA5 is the most accurate, followed by AdvHG, E3DVAR, and ERA-I.
Meanwhile, for July (Figs. 3b, d, f, and h), ERA5 shows the smallest RMSE, and AdvHG and
E3DVAR show comparable RMSE to ERA-I.

Furthermore, general features of 36 h forecast RMSE (Fig. 4) are similar to the 24 h

forecast RMSE (Fig. 3). However, particularly in January, the 36 h forecast RMSE differences
between ERA5 and ERA-I are more distinctive compared to those of 24 h forecast. In January,
the vertically averaged 36 h forecast RMSE differences of ERA5 and ERA-I are 0.52 m s$^{-1}$ for
wind, 0.16 K for temperature, and 0.08 g kg$^{-1}$ for Qvapor, whereas those of 24 h forecast are
0.4 m s$^{-1}$ for wind, 0.11 K for temperature, and 0.06 g kg$^{-1}$ for Qvapor. In addition, the 36 h
forecast RMSE differences between ERA5 and AdvHG for January are on average 0.1 m s$^{-1}$
for wind, 0.05 K for temperature, and 0.02 g kg$^{-1}$ for Qvapor, which are even smaller compared
to those of 24 h forecast, implying that AdvHG is a lot more accurate than ERA-I for January
in 2017. For July, 36 h forecast RMSE of ERA5 is the smallest and RMSEs of AdvHG and
E3DVAR are similar to those of ERA-I.
*4.1.2   RMSE and spread for the period of 2017-18*

In this section, EARR produced in this study is verified for a longer period with WRF-

based ERA5. RMSE and spread of reanalyses and reforecasts based on AdvHG method are
calculated and averaged over the period of 2017–2018. The reanalyses and (re)forecast fields
are evaluated by calculating RMSE valid at 00 and 12 UTC and spread at 00, 06, 12, and 18
UTC.

The averaged RMSEs of reanalysis for ERA5 and EARR (denoted as AdvHG in Fig. 5)

and spread of analysis and 6 h forecast fields of EARR (AdvHG) are shown in Fig. 5. With



respect to spread, the ensemble spreads of analysis fields are smaller than those of 6 h forecast
fields, on average, by 0.16 m s$^{-1}$ for wind, 0.04 K for temperature, and 0.02 g kg$^{-1}$ for Qvapor,
which is the well-known characteristics of ensemble-based data assimilation methods. To be
specific, the wind spread (Figs. 5a and b) is similar to or greater than the wind RMSE except
for the upper troposphere above 200 hPa, implying ensemble spread for wind is well
represented below 200 hPa. Even if the ensembles for temperature (Fig. 5c) are underdispersive
compared to RMSE of temperature, overall Qvapor spread (Fig. 5d) is well represented except
for 1000 hPa and above 200 hPa.

Regarding reanalysis RMSE, overall ERA5 RMSE is smaller than AdvHG RMSE for all

variables (Fig. 5). The vertically averaged RMSEs of ERA5 are smaller by 0.15 m s$^{-1}$ for wind,
0.08 K for temperature, and 0.01 g kg$^{-1}$ for Qvapor than those of AdvHG. Nonetheless, the
wind RMSEs of AdvHG are similar to those of ERA5 for the middle of troposphere (400–850
hPa), and the Qvapor RMSEs of AdvHG are similar to those of ERA5 except for 1000 hPa.

In addition, regarding 24 h forecast RMSE, ERA5 shows smaller RMSE than AdvHG for

all variables (Fig. 6). The vertically-averaged RMSE differences of wind, temperature, and
Qvapor variables between AdvHG and ERA5 are approximately 0.2 m s$^{-1}$, 0.07 K, and 0.03 g
kg$^{-1}$, respectively. These differences are smaller, compared to the 24 h forecast RMSE
difference between ERA-I and ERA5 shown in Fig. 3 (i.e., wind, Temp, and Qvapor RMSE
difference: 0.4 m s$^{-1}$, 0.11 K, and 0.06 g kg$^{-1}$ for January 2017, 0.25 m s$^{-1}$, 0.05 K, and 0.04 g
kg$^{-1}$ for July 2017).
*4.2   Evaluation of precipitation for January and July in 2017.*
*4.2.1    Evaluation metrics*
*4.2.1.1 Equitable threat score and Frequency bias index*

In this section, for the point-based Equitable threat score (ETS) and Frequency bias index

(FBI) based on Table 2, the 6 h accumulated precipitation fields based on the 6 h forecast of



E3DVAR, AdvHG, WRF-based ERA-I, WRF-based ERA5, ERA-I_fromECMWF, and
ERA5_fromECMWF are evaluated every 6 h (00, 06, 12, and 18 UTC) for January and July in
2017 (Fig. 7). Here, all the WRF-based precipitation fields are based on 12-km horizontal
resolution, and ERA-I_fromECMWF and ERA5_fromECMWF have 79- and 31-km horizontal
resolutions, respectively. Generally, ETS decreases as a threshold increases for both two
months (Figs. 7a and c). For January in 2017 (Fig. 7a), AdvHG ETS is the greatest among
others. Compared to precipitation reforecasts from ECMWF (i.e., ERA-I_fromECMWF,
ERA5_fromECMWF), AdvHG shows the higher ETS, indicating that AdvHG is able to
simulate more accurate precipitation fields than ERA-I and ERA5 from ECMWF in January
2017. Surprisingly, ETS of ERA5_fromECMWF for January in 2017 is the lowest among all
the results compared and is even lower than that of ERA-I_fromECMWF.

Since the precipitation reforecasts from ECMWF have not only coarser resolutions but

also different forecast model (i.e., the forecasting system of ECMWF), the precipitation
forecasts of ERA5 and ERA-I are additionally produced by using the same forecast model with
the same resolution as AdvHG and E3DVAR in this study, as explained in section 2.4. For
January 2017 (Fig. 7a), ETS of ERA5 (i.e., WRF-based ERA5) is higher than that of
ERA5_fromECMWF for all thresholds, whereas ETS of ERA-I (i.e., WRF-based ERA-I) is
lower than that of ERA-I_fromECMWF except for strong thresholds. The ERA5 ETS is greater
than the ERA-I ETS, but is smaller than the AdvHG ETS. The AdvHG shows the greatest ETS
among others with the same resolution and forecast model, and E3DVAR, ERA5, and ERA-I
follow.

Regarding FBI in winter (Fig. 7b), for strong thresholds, all the results show the FBI

smaller than 1, implying the underestimation of frequency of precipitation for strong thresholds.
While FBIs of ERA5_fromECMWF and ERA-I_fromECMWF are greater than 1 for weak
thresholds, those WRF-based results are similar to 1 or smaller than 1. In general, AdvHG



shows the FBI closest to 1 among all the results, which is consistent with the greatest ETS of
AdvHG. The E3DVAR FBI is similar to the AdvHG FBI, and ERA5 and ERA-I FBIs are
similar to each other. FBIs of ERA5 and ERA-I are smaller than those of AdvHG and E3DVAR.

Meanwhile, overall, the ETS values for January whose maximum is around 0.4 (Fig. 7a)

are much greater than those for July in 2017 whose maximum is around 0.2 (Fig. 7c), implying
that the precipitation forecast in summer is more difficult than that in winter. The ETS
difference between the results in July is smaller than those in January. Particularly, for the
thresholds 4 and 8 mm $(6\ h)^{-1}$, ETSs in July are similar to each other (Fig. 7c). Except for those
two thresholds, the ETS of ERA-I_fromECMWF is the smallest. At the threshold 16 mm $(6\ h)^{-1}$
, ERA5 ETS is the highest, followed by AdvHG, E3DVAR, ERA-I, ERA5_fromECMWF, and
ERA-I_fromECMWF. At the threshold 0.5 and 1 mm $(6\ h)^{-1}$, the E3DVAR ETS is the greatest,
followed by ERA5, AdvHG, ERA5_fromECMWF, ERA-I, and ERA-I_fromECMWF.

With respect to FBI in July 2017, the WRF-based results show the FBIs greater than 1,

whereas reforecast from ECMWF show the FBIs greater than 1 for weak thresholds and smaller
than 1 for strong thresholds (Fig. 7d). For July in 2017, in general, ERA5_fromECMWF FBI
is the closest to 1, followed by E3DVAR, AdvHG, ERA5, ERA-I, and ERA-I_fromECMWF
FBI.
*4.2.1.2 Probability of detection and False alarm ratio*

The Probability of Detection (POD or Hit Rate) and False Alarm Ratio (FAR) are

calculated for precipitation simulated from E3DVAR, AdvHG, WRF-based ERA-I, WRF-
based ERA5, ERA-I_fromECMWF, and ERA5_fromECMWF for January and July in 2017
(Fig. 8). For January in 2017, AdvHG POD is the greatest among the WRF-based results,
followed by E3DVAR, ERA5, and ERA-I (Figs. 8a and b). Overall, the results of reforecast
from ECMWF (i.e., ERA-I_fromECMWF and ERA5_fromECMWF) have greater POD than
the WRF-based POD for weak thresholds, whereas those have smaller POD than the WRF-



based POD for strong thresholds. Regarding FAR, notably, ERA5_fromECMWF shows
extremely great FAR and ERA5 shows the smallest FAR among all the results, which is a
consistent result with the smallest ETS of ERA5_fromECMWF. In addition to the lowest ETS
of ERA5_fromECMWF for January in 2017 as discussed in the section 4.2.1.1, FAR of
ERA5_fromECMWF is extremely high with low POD in winter. Therefore, especially for
January in 2017, the precipitation fields simulated from ERA5_fromECMWF over East Asia
are much less accurate than any other results from this study.

For July in 2017, generally, ERA5 shows the largest POD, followed by AdvHG, ERA-I,

E3DVAR, ERA5_fromECMWF (Figs. 8c and d). The ERA-I POD shows the largest POD for
weak thresholds and the smallest POD for strong thresholds, compared to other results. With
respect to FAR, FAR values in July is much greater than those in January, which is consistent
with the ETS difference between these two seasons. Overall, for strong thresholds, ERA-I
shows the highest FAR and ERA-I_fromECMWF shows the smallest FAR. For weak
thresholds, the ERA-I_fromECMWF shows the highest FAR and E3DVAR shows the smallest
FAR among all the results.
*4.2.1.3 Brier skill score*

The neighborhood sizes are chosen to be  3Δx,  5Δx,  9Δx, and  11Δx, which are 36, 60,

108, and 132 km, respectively, and the thresholds 0.5, 1, 4, 8, and 16 mm $(6\,h)^{-1}$ are considered.
The probabilistic precipitation forecasts are calculated at every model grid point depending on
neighborhood sizes and thresholds. Regarding each observation, the nearest model grid point
to observations is considered as the center of neighborhood. For verification, 6 h accumulated
precipitation fields are extracted from the first 0–6 h forecast fields of WRF-based ERA-I,
WRF-based ERA5, E3DVAR, and AdvHG every 6 h (00, 06, 12, and 18 UTC). BSSs of
ERA5_fromECMWF and ERA-I_fromECMWF are not calculated, because they have different
resolution from WRF-based results.



Based on the neighborhood approach, Brier skill score (BSS) is calculated depending on
different neighborhood sizes for January and July in 2017, respectively (Fig. 9). Because the
reference of Brier score is chosen as the ERA-I, the positive BSS implies better accuracy than
ERA-I. In general, for both two months, AdvHG BSS is greater than ERA5 BSS. Although the
E3DVAR BSS is the greatest in July 2017, the AdvHG BSS is the greatest in January 2017.
For January in 2017, as a neighborhood size increases, AdvHG and E3DVAR BSSs tend
to increase except for ERA5. Overall, AdvHG BSS is the greatest among other BSSs for all
thresholds for all neighborhood sizes. The ERA5 BSS is greater than E3DVAR BSS except for
16 mm (6 h)$^{-1}$. The highest BSS of AdvHG and the lowest BSS of ERA-I are consistent with
ETS result. Unlike greater E3DVAR ETS than ERA5 ETS, ERA5 BSS is greater than E3DVAR
BSS in January 2017.
For July 2017, while the ETS difference between the WRF-based results is not distinct
(Fig. 7c), the BSS difference is rather noticeable. Generally, E3DVAR BSS is the greatest
among other BSSs for all thresholds except for 16 mm (6 h)$^{-1}$ for neighborhood sizes 9 and 11.
Although E3DVAR BSS is the largest, AdvHG outperforms ERA5 and ERA-I. The worst
performance of ERA-I precipitation is consistent with ETS result. At weak thresholds,
E3DVAR BSS is the greatest, which is similar to ETS. For strong thresholds, ERA5 ETS is the
highest, followed by AdvHG and E3DVAR, whereas overall E3DVAR BSS is the highest,
followed by AdvHG and ERA5.
*4.2.2   Spatial distribution*
*4.2.2.1   6 h accumulated precipitation with the pattern correlation coefficient*
In this section, the spatial distributions of 6 h accumulated precipitation from the WRF-
based forecast and reforecast from ECMWF are compared. In addition, pattern correlation
coefficients (PCC) are calculated and shown at the bottom right of Figs. 10 and 11.
The PCC is computed according to the usual Pearson correlation operating on the N



observed point pairs of 6 h accumulated precipitation fields simulated from (re)forecast and
observations at the specific time. For the calculation of PCC, 6 h accumulated precipitation
fields from (re)forecast fields are interpolated bilinearly to the N observed points.

Firstly, on 29th and 30th of January in 2017 (Fig. 10), it is noticeable that the precipitation

of ERA5_fromECMWF does not match observations well over East Asia compared to other
simulated precipitation fields. As shown in Fig. 10g, ERA5_fromECMWF incorrectly
simulates precipitation over South East China, whereas other results do not forecast
precipitation over this area. In addition, ERA5_fromECMWF overestimates precipitation over
inland area of China (Fig. 10zz), whereas other results simulate precipitation similar to
observations regarding its position and intensity. ERA5_fromECMWF also shows noticeably
smaller PCC (Figs. 10g, n, and zz). Although PCC does not represent the exact accuracy or
predictability of precipitation, the overall feature of PCC is consistent with the results found so
far. In particular, PCCs of ERA5_fromECMWF are much smaller than those of other
precipitation fields. For January in 2017, the averaged PCC of AdvHG is the greatest (i.e., 0.61)
and that of ERA5_fromECMWF is the smallest (i.e., 0.46) (not shown).

Secondly, for 1st and 2nd of July in 2017 (Fig. 11), overall, the precipitation simulated from

ERA5_fromECMWF is well represented, compared to January in 2017 shown in Fig. 10. The
ERA-I_fromECMWF fails to simulate heavy rain for summer season due to its coarse
resolution. Furthermore, during July in 2017, ERA5 and ERA-I simulate heavier precipitation
than AdvHG (not shown), which is consistent with larger FBI of ERA5 and ERA-I at strong
thresholds. For one-month period of July in 2017, the averaged PCC of ERA5 is the greatest
(i.e., 0.37) and that of AdvHG is 0.34, but the PCC difference between ERA5 and AdvHG is
not distinctive. Moreover, the overall range of averaged PCC of different datasets in summer
(i.e., 0.29-0.35) is smaller than that in winter (i.e., 0.46-0.61), which is consistent with the
seasonal difference of ETS in this study.





*4.2.2.2   Monthly accumulated precipitation*
In this section, the monthly accumulated precipitation fields of rain gauge based
observations, E3DVAR, AdvHG, ERA-I, ERA5, ERA-I_fromECMWF, and
ERA5_fromECMWF are compared to each other for two one-month periods in January and
July in 2017, respectively.
Although all the results similarly represent overall features of precipitation in January (Fig.
12), ERA5_fromECMWF (Fig. 12g) simulates the overestimated precipitation over South
China, compared to other results and observations, which is consistent with the results in the
previous section as well as its larger FBI at weak thresholds shown in Fig. 7b. It is noticeable
that all results fail to represent the observed precipitation area over Tibetan Plateau (25°–40°N,
95–105°E). The monthly accumulated precipitation fields simulated by E3DVAR and AdvHG
(Figs. 12 b and c) are similar to each other, and E3DVAR and AdvHG produce the best fit to
observed fields. Especially, for the north-western part of Japan (e.g., Chugoku and Kinki),
E3DVAR and AdvHG are able to represent precipitation correctly, whereas ERA-
I_fromECMWF and ERA5_fromECMWF fail to do so (Fig. 12).
For the monthly accumulated precipitation in July 2017, overall, the ERA5_fromECMWF
(Fig. 13g) and the WRF-based results (Fig. 13b, c, and e) except for ERA-I (Fig. 13d) well
simulate precipitation similar to observations. ERA-I_fromECMWF is not able to simulate
heavy precipitation over Korea. For western and southern part of Japan, while ERA-
I_fromECMWF and ERA5_fromECMWF simulate similar precipitation fields to observed
fields, WRF-based results overestimate precipitation over these regions. Compared to ERA-
I_fromECMWF and ERA5_fromECMWF, the WRF-based results tend to overestimate
precipitation in South China, Korea, and Japan. This is consistent with the result in Fig. 7d, in
which FBIs from WRF-based results are generally greater than 1 for strong thresholds, whereas
those from ECMWF are smaller than 1.



Even though detailed precipitation features of WRF-based results are different, overall
features of precipitation from WRF-based results are similar to each other, which implies that
predictability of precipitation strongly depend on the physics schemes as well as NWP model,
especially for summer season. According to Que et al. (2016), depending on the combinations
of physics options in WRF model, the spatial distribution of precipitation can be significantly
different over Asian summer monsoon area and YSU PBL scheme which is used in this study
tends to overestimate precipitation over the same area. Thus, different physics options could
simulate the different spatial distribution of precipitation.
In addition, compared to ERA5 based on WRF model (Fig. 13e), ECMWF model for
ERA5_fromECMWF (Fig. 13g) seems to suppress precipitation. Thus, WRF model with the
physics schemes used in this study might simulate more precipitation than ECMWF model,
although the initial condition is the same. Therefore, it is important to consider the consistency
of the systems for data assimilation and forecast model for a good performance of precipitation.
## 5. Data Availability
The EARR data presented in this study are available every 6 h (i.e., 00, 06, 12, and 18
UTC)    for    the    period    of    2010-2019    from    Harvard    Dataverse    Repository
(https://dataverse.harvard.edu/dataverse/EARR). The EARR 6 hourly data on pressure levels
(https://doi.org/10.7910/DVN/7P8MZT, Yang and Kim 2021b) and 6 hourly precipitation data
(https://doi.org/10.7910/DVN/Q07VRC, Yang and Kim 2021c) are provided in NetCDF file
format.
The EARR 6 hourly data on pressure levels (Yang and Kim 2021b) include u-component
of wind, v-component of wind, temperature, geopotential height, and specific humidity
variables of reanalysis on pressure levels (i.e., 925, 850, 700, 500, 300, 200, 100, and 50 hPa).
The EARR 6 hourly precipitation data (Yang and Kim 2021c) contain 6 h accumulated total



precipitation variable of 6 h reforecast on single level. The 6 h accumulated total precipitation
is obtained from 6 h reforecast field which is integrated for 6 h from reanalysis field every 6 h
(i.e., 00, 06, 12 and 18 UTC).

## 484    6. Summary and conclusions

In this study, to develop the regional reanalysis system over East Asia, the advanced
hybrid gain algorithm (AdvHG) is newly proposed and evaluated with traditional hybrid DA
method (E3DVAR) as well as existing reanalyses from ECMWF (ERA5 and ERA-I) for
January and July in 2017. The East Asia Regional Reanalysis (EARR) system is developed
based on the AdvHG as the data assimilation method using WRF model and conventional
observations, and the high-resolution regional reanalysis and reforecast fields with 12 km
horizontal resolution are produced over East Asia for the ten-year period of 2010–2019.
The AdvHG newly proposed in this study is based on the hybrid gain approach, weighting
analysis from variational-based and ensemble-based DA algorithms to generate optimal hybrid
analysis, which can play an important role as a simple and practical method in the foreseeable
future to take advantage of each strength of two different methods. The advanced hybrid gain
method is different from the hybrid gain approach in that 1) E3DVAR is used instead of EnKF,
2) 6 h forecast of ERA5 is used instead of deterministic analysis for a more balanced and
consistent analysis with WRF model, and 3) the pre-existing and state-of-the-art reanalysis data
(i.e. ERA5) is simply used instead of producing our own analysis fields from a variational DA
method. Thus, it can be regarded as an efficient approach to generate regional reanalysis dataset
because of cost savings as well as the use of the state-of-the-art reanalysis from ECMWF that
assimilates all available observations.
For a verification, the latest ECMWF reanalysis and reforecast datasets (i.e., ERA5 and
ERA-I) are used. With respect to forecast variables, two different forecast fields of ECWMF



are used: 1) reforecast fields from ECMWF (i.e., ERA5_fromECMWF and ERA-
I_fromECMWF) and 2) forecast fields (i.e., WRF-based ERA5 and WRF-based ERA-I)
integrated in WRF model with 12 km resolution using ERA5 and ERA-I as initial conditions.
To evaluate this newly proposed algorithm, analysis and forecast wind, temperature, and
humidity variables are evaluated with respect to RMSE and spread for January and July in 2017.
Overall, the analysis RMSE of E3DVAR is the smallest among others but comparable to that
of ERA5, especially for January. Regarding forecast variables, AdvHG outperforms E3DVAR
and ERA5 outperforms ERA-I for January and July in 2017. Although ERA5 outperforms
AdvHG for upper air variables for two seasons, AdvHG outperforms ERA-I in January and
shows comparable performance to ERA-I in July. Additionally the verification results of
AdvHG and ERA5 for the period of 2017-18 are consistent with those for two one-month
period in 2017.
The precipitation forecast variables are also verified regarding a neighborhood-based
verification score (i.e., Brier skill score) as well as the point-based verification scores (i.e., ETS,
FBI, POD, and FAR). According to the point-based verification scores, the precipitation
forecast of AdvHG in January is the most accurate, followed by E3DVAR, ERA5, ERA-I. The
precipitation reforecast of ERA5_fromECMWF shows the worst performance with the lowest
ETS and the highest FAR among other results in January. For July, overall ETS values of all
results are relatively lower compared to those in January, implying the lower predictability in
summer season. For July, ERA5 shows the greatest ETS for strong thresholds followed by
AdvHG and E3DVAR, and E3DVAR ETS is the greatest followed by ERA5 and AdvHG for
weak thresholds. However, the ETS differences between the results are not distinctive.
To prevent from double penalty when verifying a highly variable data with high resolution
(e.g., precipitation), Brier skill score (BSS) based on neighborhood approach is calculated for
6 h accumulated precipitation forecasts depending on different neighborhood sizes for January



and July in 2017. In general, BSS of AdvHG is greater than that of ERA5 and ERA-I for both
two months. Although the E3DVAR BSS is the greatest in July 2017, the AdvHG BSS is the
greatest in January 2017.

Lastly, the spatial distributions of 6 h and monthly accumulated precipitation forecast for

AdvHG, E3DVAR, ERA-I, ERA5, ERA-I_fromECMWF, and ERA5_fromECMWF are
compared with rain-gauge based observations. For January 2017, it is noticeable that AdvHG
precipitation is the closest to observations with highest PCC (i.e., 0.61) and
ERA5_fromECMWF overestimates precipitation over South China with the lowest PCC (i.e.,
0.46). For July in 2017, due to a coarse resolution of ERA-I_fromECMWF, it fails to represent
heavy rain over East Asia. Meanwhile, the WRF-based results tend to overestimate
precipitation compared to ERA-I_fromECMWF and ERA5_fromECMWF. In addition, even
though the averaged PCC of ERA5 (i.e., 0.37) is slightly greater than that of AdvHG (i.e., 0.34),
the PCC difference between ERA5 and AdvHG is not distinctive and overall range of averaged
PCC of all datasets in summer (i.e., 0.29-0.35) is smaller than that in winter (i.e., 0.46-0.6).

In conclusion, for upper air variables, overall, ERA5 outperforms EARR based on AdvHG,

but the RMSE difference between ERA5 and EARR (AdvHG) is smaller than that between
ERA5 and ERA-I. In addition, EARR outperforms ERA-I for January 2017 and shows
comparable performance to ERA-I for July 2017. On the contrary, according to the evaluation
results of precipitation, in general, EARR better represents precipitation than ERA5 as well as
ERA5_fromECMWF for January and July in 2017. Even if E3DVAR precipitation is better
represented than EARR precipitation for July, the difference is not considerable for July and
EARR better simulates precipitation for January than E3DVAR. Therefore, although the
uncertainties of upper air variables of EARR should be considered when analyzing them, the
precipitation reforecast of EARR is more accurate than that of ERA5 for both two seasons.



**Author contribution**

Hyun Mee Kim proposed the main scientific ideas and Eun-Gyeong Yang contributed the supplementary ideas during the process. Eun-Gyeong Yang developed the reanalysis system and produced the 10-year regional reanalysis data. Eun-Gyeong Yang and Hyun Mee Kim analyzed the simulation results and completed the manuscript. Dae-Hui Kim contributed to analyzing the reanalysis data and to the preparation of software and computing resources for the reanalysis system.

**Competing interests**

The authors declare that they have no competing interests.

**Acknowledgments**

This study was supported by a National Research Foundation of Korea (NRF) grant funded by the South Korean government (Ministry of Science and ICT) (Grant 2017R1E1A1A03070968 and Grant 2021R1A2C1012572) and the Yonsei Signature Research Cluster Program of 2021 (2021-22-0003). This study was carried out by utilizing the supercomputer system supported by the National Center for Meteorological Supercomputer of Korea Meteorological Administration and Korea Research Environment Open NETwork (KREONET) provided by the Korea Institute of Science and Technology Information. The authors gratefully acknowledge the late Dr. Fuqing Zhang for collaborations at the earlier stages of this study.



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





**Table caption**
Table 1. Model configuration
Table 2. The $2 \times 2$ contingency table for dichotomous (yes-no) events.



**Figure caption**
Figure 1. The model domain over East Asia with verification area (black dashed box).
Figure 2. RMSEs of analysis of (a,b) zonal wind, (c,d) meridional wind, (e,f) temperature, and
(g,h) Qvapor (water vapor mixing ratio) from ERA-I (black dashed), ERA5 (black solid),
E3DVAR (blue dashed), AdvHG (blue solid) depending on pressure levels for (left) January
and (right) July in 2017.
Figure 3. Same as Fig. 2 except for 24 h forecast.
Figure 4. Same as Fig. 2 except for 36 h forecast.
Figure 5. RMSEs of analysis of (a) zonal wind, (b) meridional wind, (c) temperature, and (d)
Qvapor (water vapor mixing ratio) from ERA5 (black solid) and AdvHG (blue solid) and
spreads of analysis (black dashed) and 6 h forecast (gray dashed) of AdvHG depending on
pressure levels averaged over the two-year period of 2017–2018.
Figure 6. Same as Fig. 5 except for RMSE of 24 h forecast.
Figure 7. (a,c) ETS and (b,d) FBI for (a,b) January and (c,d) July in 2017 depending on
thresholds 0.5, 1, 4, 8, and 16 mm (6 h)$^{-1}$.
Figure 8. (a,c) POD and (b,d) FAR for (a,b) January and (c,d) July in 2017 depending on
thresholds 0.5, 1, 4, 8, and 16 mm (6 h)$^{-1}$.
Figure 9. Brier skill score of the probabilistic postprocessed forecast with reference to the
WRF-based ERA-I for (a-d) January and (e-h) July in 2017 (Blue solid: AdvHG, blue dashed:
E3DVAR, red solid: WRF-based ERA5).
Figure 10. The spatial distribution of 6 h accumulated precipitation of (1st column) observation,
(2nd column) E3DVAR, (3rd column) AdvHG, (4th column) ERA-I, (5th column) ERA5, (6th



column) ERA-I_fromECMWF, and (7th column) ERA5_fromECMWF and the pattern
correlation coefficient (PCC) shown at the bottom right of each figure at valid time (1st low, 3rd
low) 06 UTC and (2nd low, 4th low) 18 UTC on 29th and 30th of January in 2017.
Figure 11. As in Fig. 10, but for 1st and 2nd of July in 2017.
Figure 12. The spatial distribution of the monthly accumulated precipitation of (a) observations,
(b) E3DVAR, (c) AdvHG, (d) ERA-I, (e) ERA5, (f) ERA-I from ECMWF, and (g) ERA5 from
ECMWF for January 2017.
Figure 13. As in Fig. 12, but for July 2017.



Table 1. Model configuration

| | Description |
|---|---|
| **Hori. Resol.** | 12 km (540×432 grid points) |
| **Vert. Lev.** | 50 vertical levels (up to 5 hPa) |
| **Model** | WRF Model (v3.7.1, Skamarock et al. 2008) |
| **LBC** | ERA5 (Hersbach et al. 2020) |
| **Data assimilation** | E3DVAR (Zhang et al. 2013), Adanced hybrid gain method |
| **Microphysics** | Thompson scheme (Thompson et al. 2008) |
| **Cumulus convection** | Grell–Freitas ensemble scheme (Grell and Freitas 2014) |
| **PBL** | Yonsei University scheme (Hong et al. 2006) |
| **Radiation** | Rapid Radiative Transfer Model (RRTMG) scheme (Iacono et al. 2008) |
| **Surface layer** | Revised MM5 Monin-Obukhov scheme (Jiménez et al. 2012) |
| **Surface model** | Unified Noah Land Surface Model (Tewari et al. 2004) |







Table 2. The  $2 \times 2$  contingency table for dichotomous (yes-no) events.

| Forecast | Observed | | |
|---|---|---|---|
| | Yes | No | |
| Yes | Hits (A) | False alarms (B) | A + B |
| No | Misses (C) | Correct rejections (D) | C + D |
| | A + C | B + D | Total = A + B + C + D |



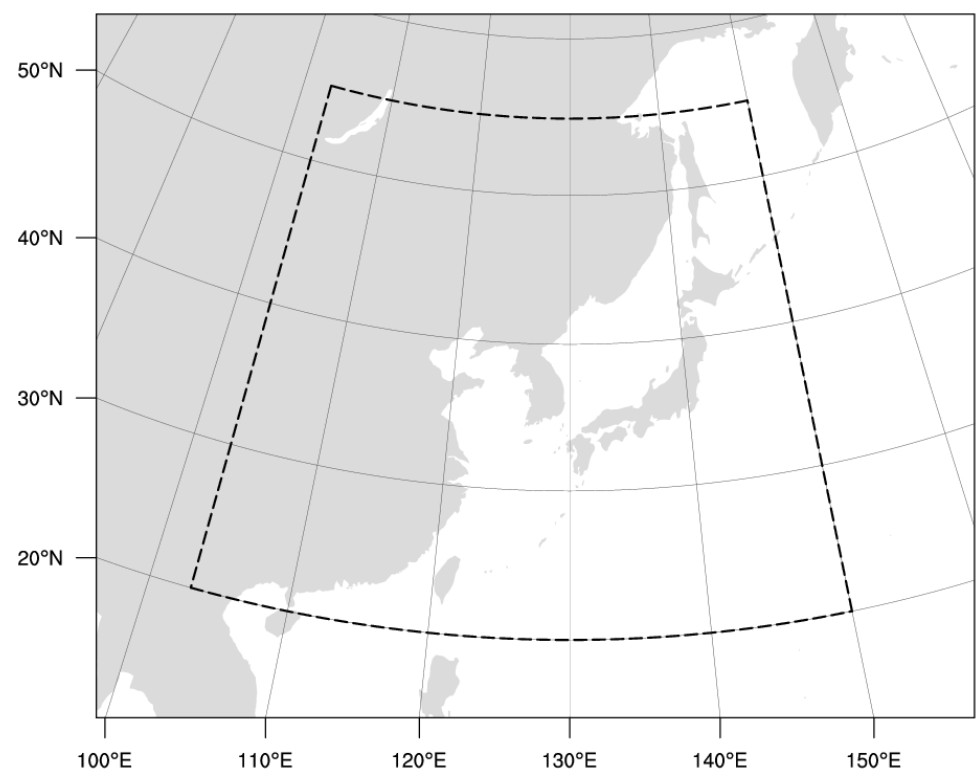


Figure 1. The model domain over East Asia with verification area (black dashed box).

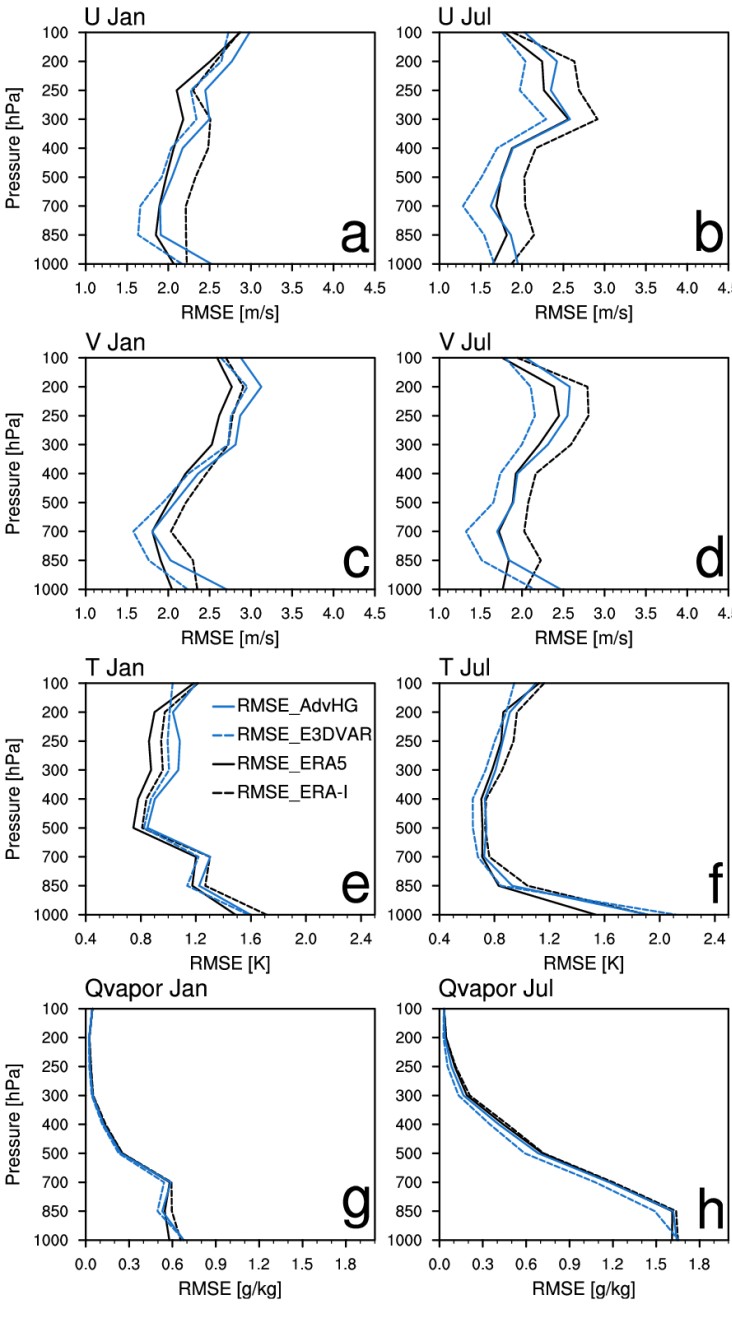


Figure 2. RMSEs of analysis of (a,b) zonal wind, (c,d) meridional wind, (e,f) temperature, and
(g,h) Qvapor (water vapor mixing ratio) from ERA-I (black dashed), ERA5 (black solid),
E3DVAR (blue dashed), AdvHG (blue solid) depending on pressure levels for (left) January
and (right) July in 2017.

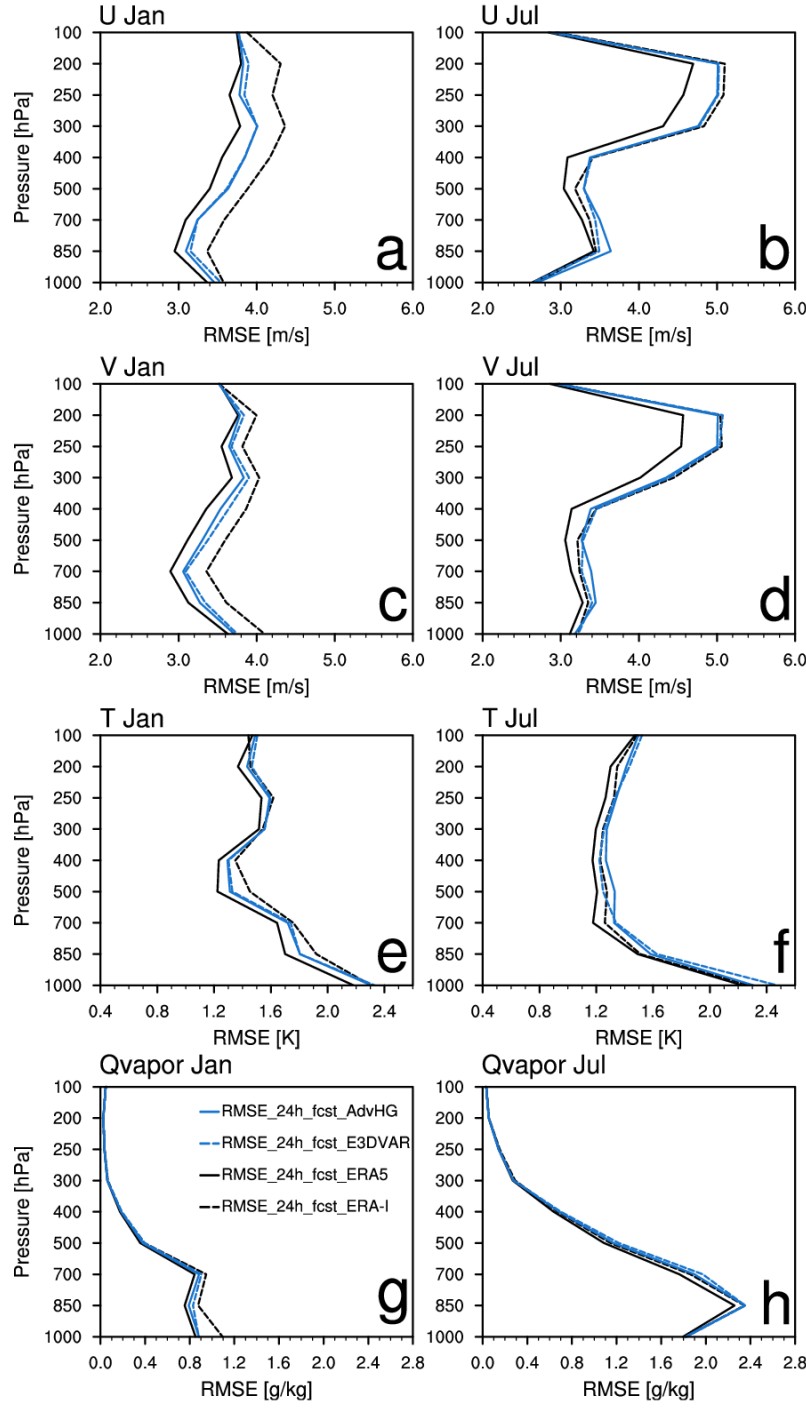


Figure 3. Same as Fig. 2 except for 24 h forecast.

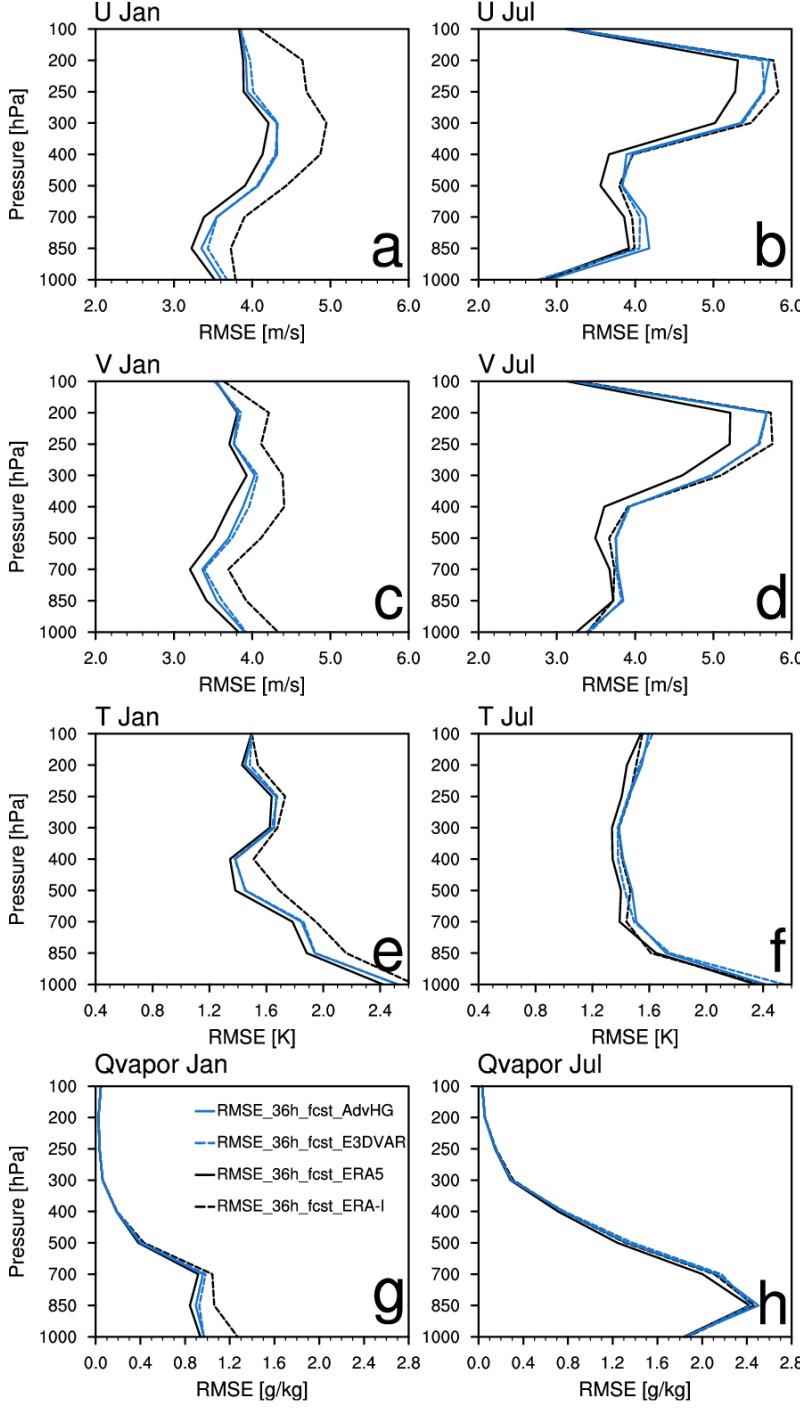


Figure 4. Same as Fig. 2 except for 36 h forecast.

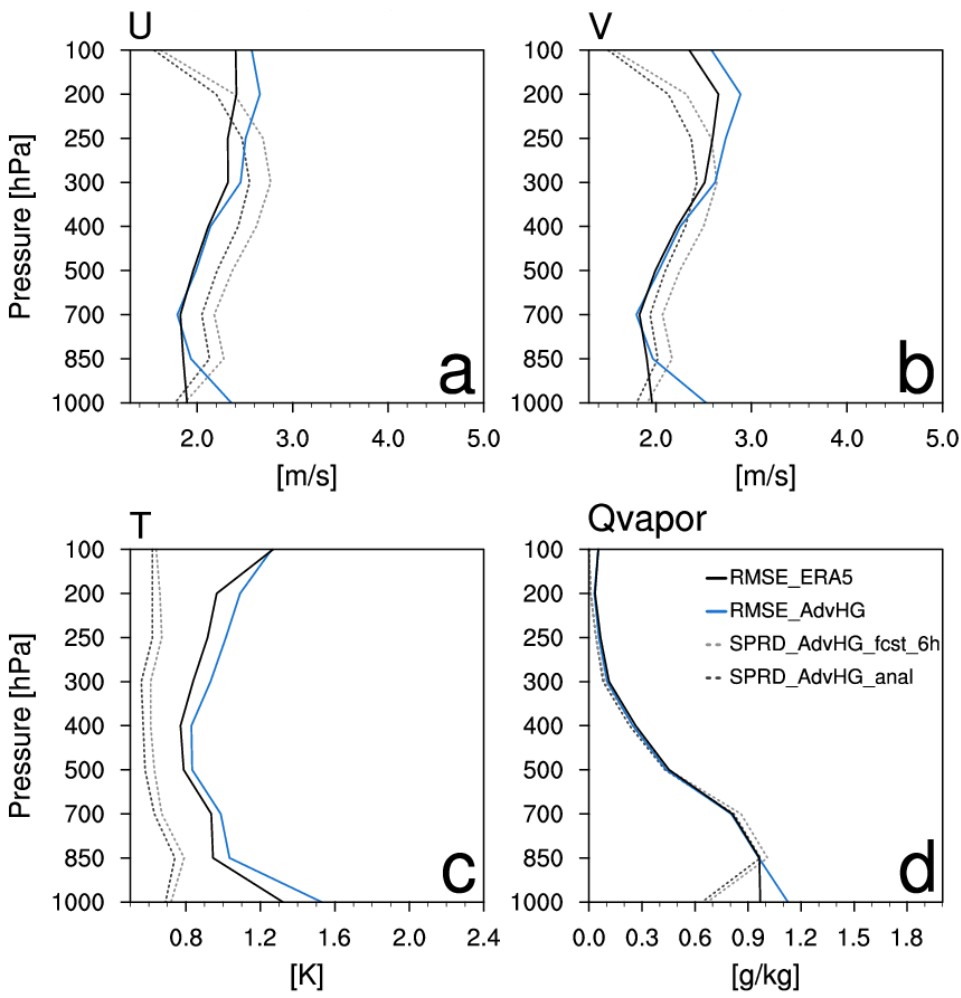

Figure 5. RMSEs of analysis of (a) zonal wind, (b) meridional wind, (c) temperature, and (d) Qvapor (water vapor mixing ratio) from ERA5 (black solid) and AdvHG (blue solid) and spreads of analysis (black dashed) and 6 h forecast (gray dashed) of AdvHG depending on pressure levels averaged over the two-year period of 2017–2018.

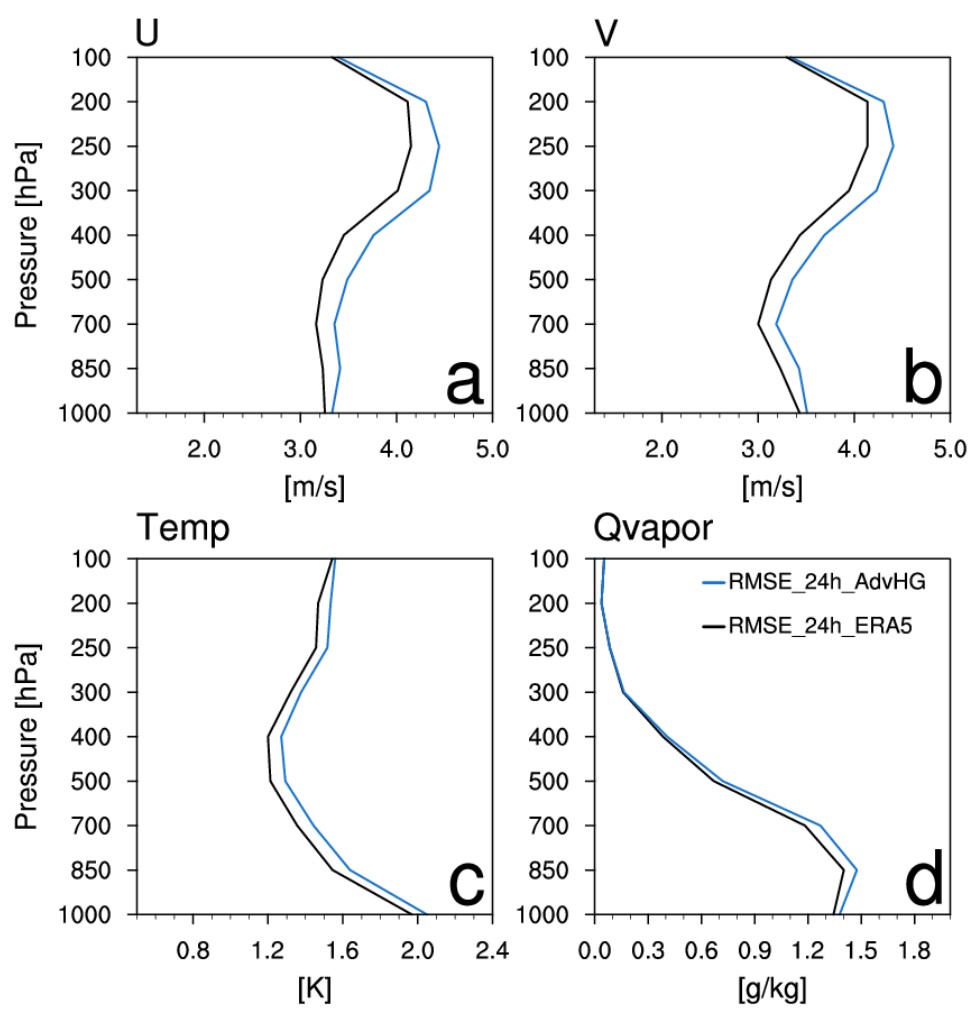


Figure 6. Same as Fig. 5 except for RMSE of 24 h forecast.



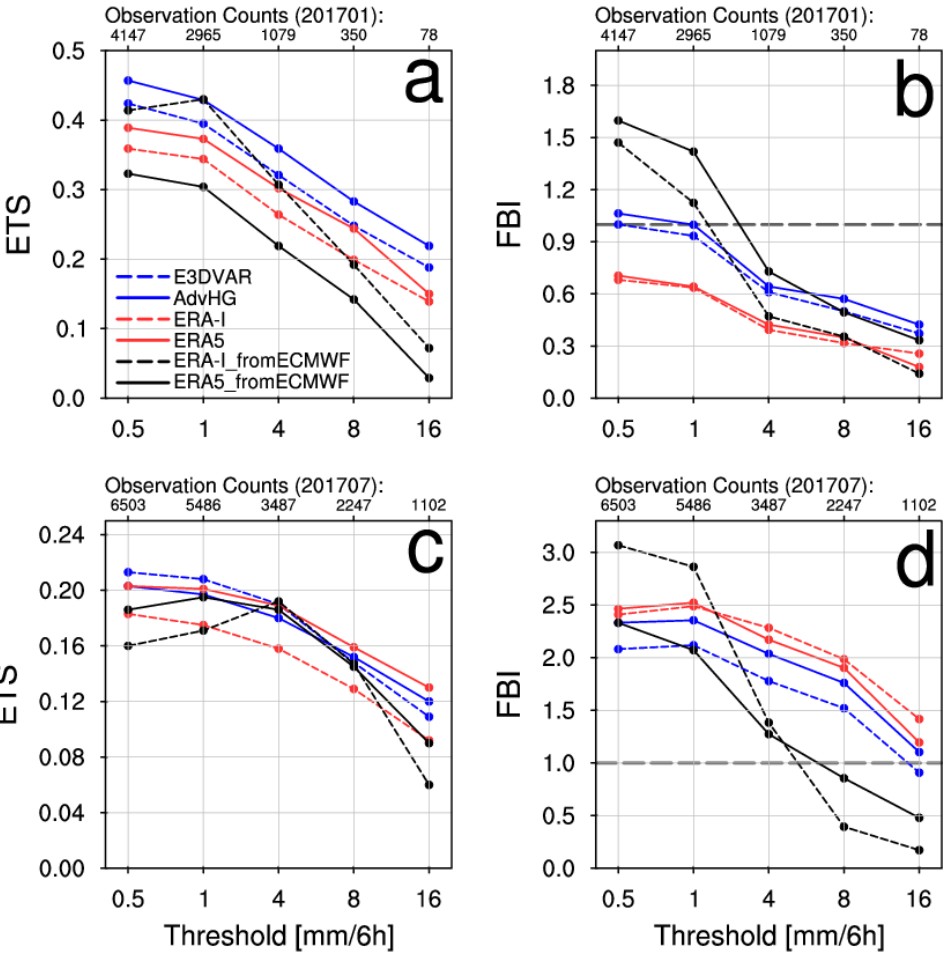


Figure 7. (a,c) ETS and (b,d) FBI for (a,b) January and (c,d) July in 2017 depending on thresholds 0.5, 1, 4, 8, and 16 mm (6 h)$^{-1}$.


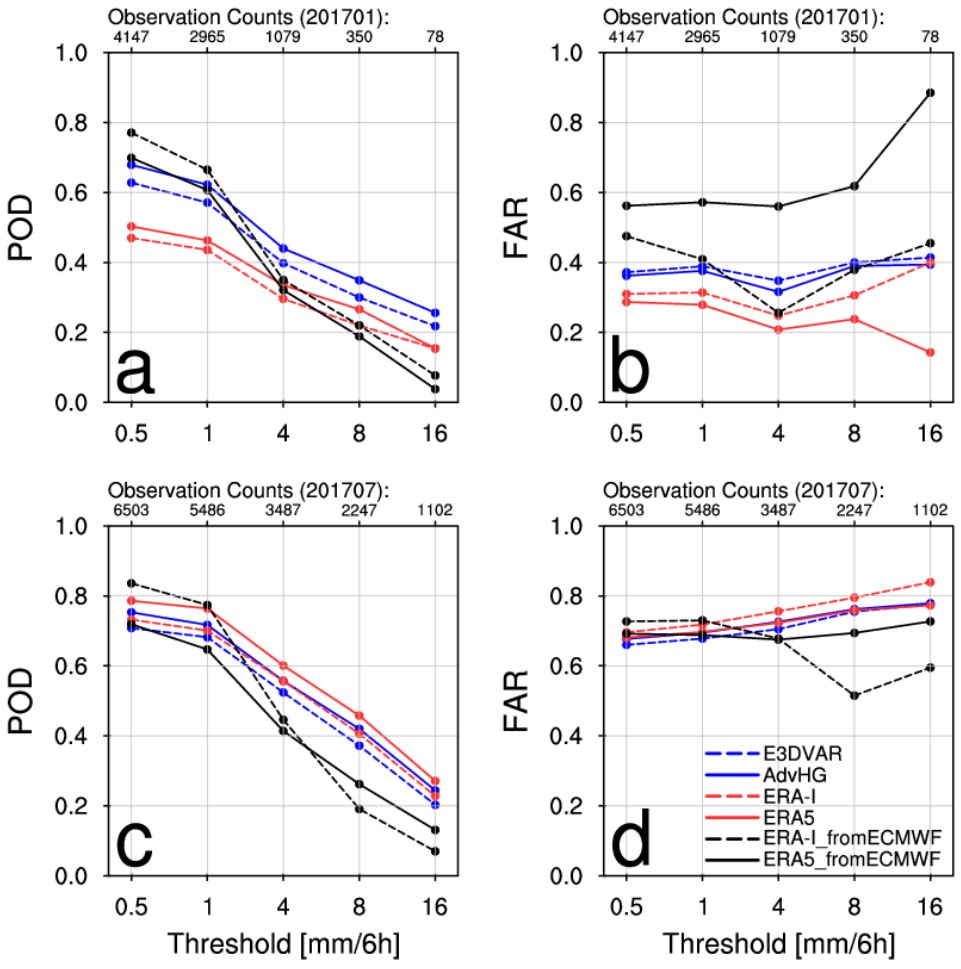


Figure 8. (a,c) POD and (b,d) FAR for (a,b) January and (c,d) July in 2017 depending on thresholds 0.5, 1, 4, 8, and 16 mm (6 h)$^{-1}$.


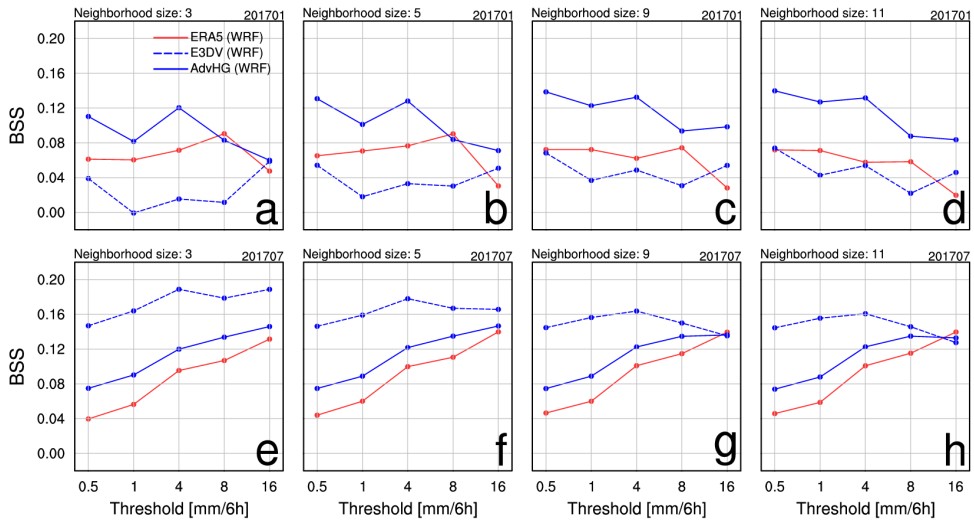

Figure 9. Brier skill score of the probabilistic postprocessed forecast with reference to the WRF-based ERA-I for (a-d) January and (e-h) July in 2017 (Blue solid: AdvHG, blue dashed: E3DVAR, red solid: WRF-based ERA5).



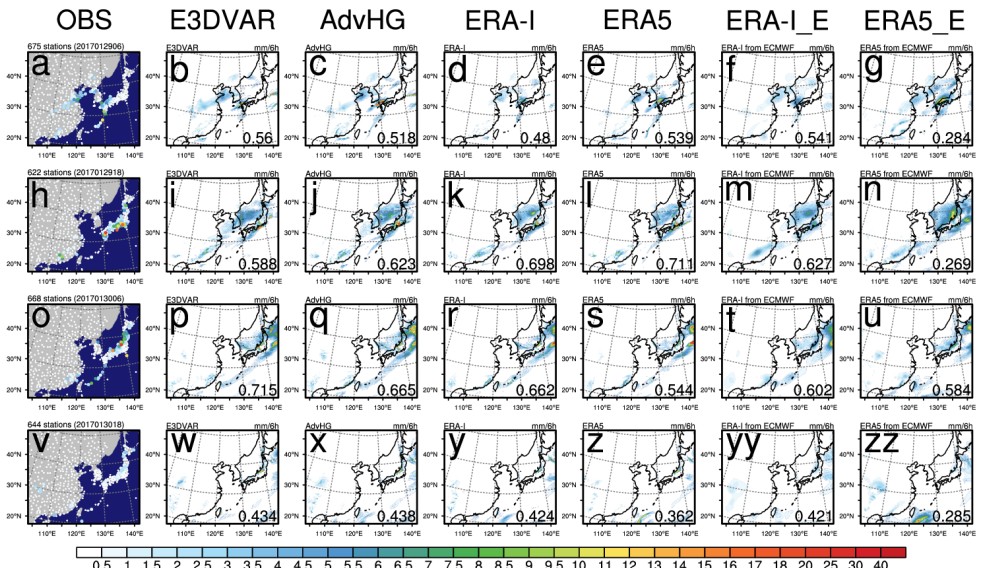

Figure 10. The spatial distribution of 6 h accumulated precipitation of (1st column) observation, (2nd column) E3DVAR, (3rd column) AdvHG, (4th column) ERA-I, (5th column) ERA5, (6th column) ERA-I_fromECMWF, and (7th column) ERA5_fromECMWF and the pattern correlation coefficient (PCC) shown at the bottom right of each figure at valid time (1st low, 3rd low) 06 UTC and (2nd low, 4th low) 18 UTC on 29th and 30th of January in 2017.





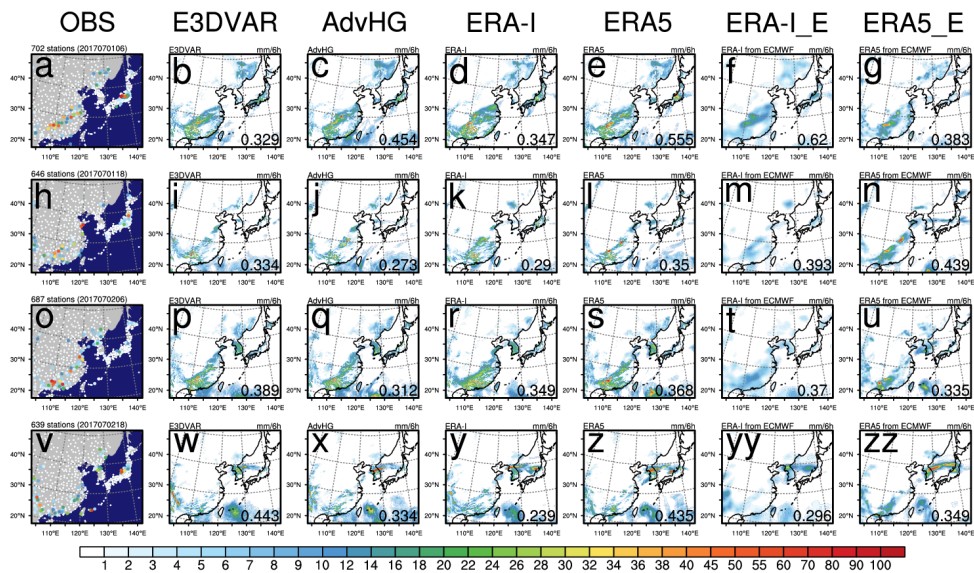

Figure 11. As in Fig. 10, but for 1st and 2nd of July in 2017.

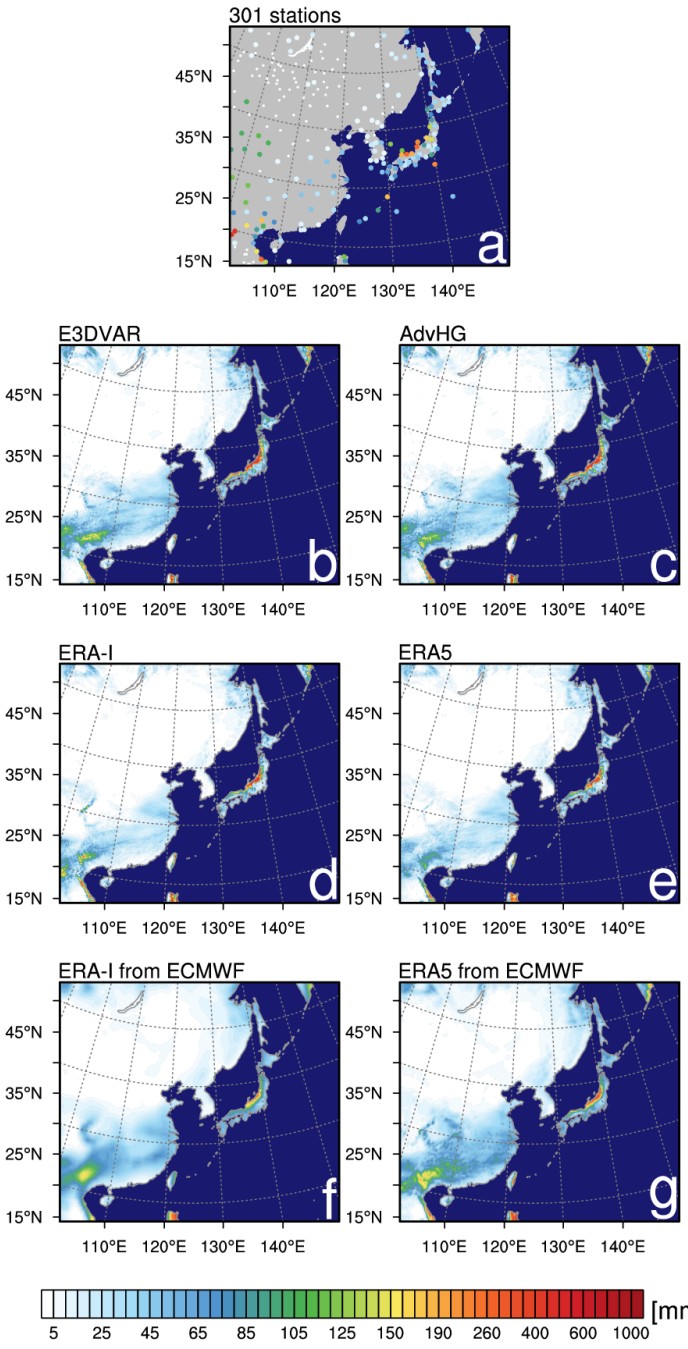

Figure 12. The spatial distribution of the monthly accumulated precipitation of (a) observations,
(b) E3DVAR, (c) AdvHG, (d) ERA-I, (e) ERA5, (f) ERA-I from ECMWF, and (g) ERA5 from
ECMWF for January 2017.

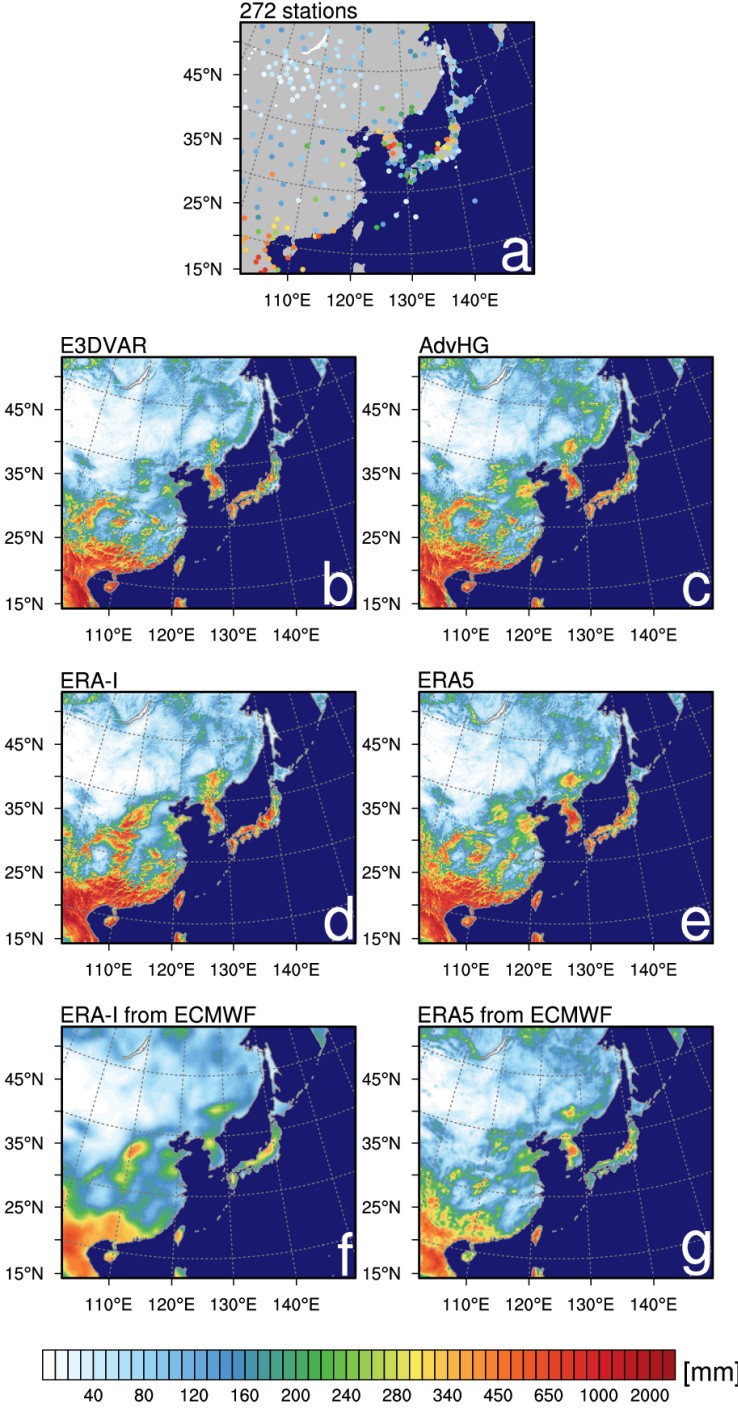


Figure 13. As in Fig. 12, but for July 2017.