# Peer review of "Development of East Asia Regional"

_Earth System Science Data, 2021_

## Author Comment (AC1)

ESSD-2021-217
Response to RC1 (Referee 1)

The authors thank referee 1 for a thoughtful review of the manuscript. We agree with many of the referee's points and have made the necessary changes. The responses for the referee's specific comments are as follows.

**Overall comments:**

*The authors generated the East Asia Regional Reanalysis (EARR) using Weather Research and Forecasting (WRF, v3.7.1) during the ten-year period 2010-2019, based on the advanced hybrid gain data assimilation method (AdvHG). The new advanced hybrid gain (AdvHG) data assimilation method combining E3DVAR and ERA5 based on WRF model is newly proposed and investigated in this study. The manuscript verified the EARR for two-year period 2017-2018 by comparing EARR against ERA-Interim, ERA5 and observations.*

*I think a lot of work behind this manuscript is worth publishing. However, there are some details and the underlying physics processes were not well discussed.*

*The detailed comments are listed as below.*

**Major comments:**

*1-A. It is not easy to understand the regional reanalysis method used in this paper. I'd suggest the authors use schematic diagram to clearly describe the regional reanalysis method.*

**Authors' response**: Following the referee's suggestion, a schematic diagram for the advanced hybrid gain method is added to the revised manuscript as Fig. 2 (Figure_rev1. below).

[Figure]

Figure_rev1. The schematic diagram of the advanced hybrid gain data assimilation method in the East Asia regional reanalysis system.

*1-B. What does "forecast fields are integrated up to 36h" mean? Is reinitialization used? What does "6h forecast of ERA5 reanalysis based on WRF model" mean? Does it mean WRF simulations using ERA5 as LBC forcing?*

**Authors' response**: "Forecast fields are integrated up to 36 h" implies that forecast fields are produced using WRF model for 36 h. Thus, 24 and 36 h forecast fields are evaluated in this study (Figs. 4, 5, and 7 in the revised manuscript).

Reinitialization is not used in this study. For the experiments using data assimilation methods based on WRF model (E3DVAR, AdvHG), analysis fields are produced every 6 h (00, 06, 12, 18 UTC) via 6-h assimilation cycle, so forecast fields are generated from these cycled analysis fields used as initial conditions. For (WRF-based) ERA5 and ERA-I experiments, 36 h forecast fields are generated using WRF model with ERA5 and ERA-I reanalysis fields used as initial conditions, so reinitialization is not required in this study.

"6 h forecast of ERA5 reanalysis based on WRF model" means that 6 h forecast fields are produced using WRF model with ERA5 reanalysis as the initial condition. For WRF-based

ERA5 experiment, ERA5 is used as the lateral boundary condition as well as the initial condition in WRF model.

*1-C. How many experiments are used in this manuscript? What experiments do "E3DVAR", "AdvHG", "WRF-ERA5" and "WRF-ERAIN" refer to respectively? These details should be more clarified.*

**Authors' response**: In this study, E3DVAR and AdvHG imply analysis or forecast fields of experiments based on WRF model using E3DVAR and AdvHG data assimilation methods, respectively. Regarding reanalysis and (re)forecast fields of ECMWF, reanalysis fields (ERA5 and ERA-I) downloaded from ECMWF are evaluated (Figs. 3 and 6 in the revised manuscript). There are two different (re)forecast fields (e.g., ERA5_fromECMWF, WRF-based ERA5) used in this study, as mentioned in section 2.4. WRF-based ERA5 and ERA-I are forecast fields based on WRF model where ERA5 and ERA-I are used as initial conditions, respectively. In contrast, ERA5_fromECMWF and ERA-I_fromECMWF are reforecast fields based on ECMWF model not WRF model, and these reforecast fields are only evaluated for precipitation (Figs. 8 and 9 in the revised manuscript). Following the referee's suggestion, detailed information on each experiment is added to section 2.4 and Table 3 in the revised manuscript (L210-219) as follows (underlined).

(L210-219) "In this study, (re)forecast as well as reanalysis fields need to be verified. Regarding reanalysis and (re)forecast fields of ECMWF, reanalysis fields (i.e., ERA5 and ERA-I) downloaded from ECMWF are evaluated (Figs. 3 and 6). There are two different (re)forecast fields (e.g., ERA5_fromECMWF, WRF-based ERA5) used in this study. WRF-based ERA5 and ERA-I are forecast fields based on WRF model with 12 km horizontal resolution where ERA5 and ERA-I are used as initial conditions, respectively. In contrast, ERA5_fromECMWF and ERA-I_fromECMWF are reforecast fields based on ECMWF model not WRF model, so the reforecast fields of ERA5 and ERA-I are provided and downloaded from ECMWF. These reforecast fields are only used for evaluation of precipitation (Figs. 8 and 9). The (re)analysis and (re)forecast fields and corresponding experiments are explained in Table 3"

*2. The newly advanced hybrid gain (AdvHG) data assimilation method uses 6h forecast ERA5 instead of deterministic analysis, and uses ERA5 instead of producing their own analysis fields from a variational DA method. The method is expected to save time and computing cost compared to traditional data assimilation framework. Why does the author use the 6h forecast of ERA5 reanalysis based on WRF instead of the deterministic analysis in AdvHG? Will the regional reanalysis be more accurate if using ERA5 deterministic analysis in AdvHG?*

**Authors' response**: In this study, 6 h forecast of ERA5 based on WRF model is used instead of ERA5 reanalysis fields to maintain the consistency between different modeling systems as well as different resolutions. ERA5 reanalysis fields are generated based on the Integrated Forecasting System (IFS) of ECMWF with approximately 30 km horizontal

resolution. If ERA5 reanalysis is directly used to combine with E3DVAR analysis based on WRF model whose horizontal resolution is 12 km, there could be some imbalance occurred in meteorological fields resulting from two different modeling systems. Thus, to reduce the imbalance and ensure the stability and consistency during analysis process, 6 h forecast of ERA5 is used instead of ERA5 deterministic reanalysis in Advanced Hybrid Gain method.

*3. EARR is developed during the ten-year period 2010-2019, while in the manuscript, the authors only verified the product for two-year period 2017-2018. Why is this two-year period instead of the entire ten-year period chosen to be verified? I would like to see how the EARR product performs compared to the other two ECMWF global reanalysis in a longer time period, and whether EARR is suitable for application in long-term climatology research. If so, the influence of this product will be greatly increased.*

**Authors' response**: One of the reasons for verifying only for two-year period in this study is that producing 36 h forecast fields of ERA5 using WRF model for 10-year period is too computationally expensive. In this study, (re)forecast as well as (re)analysis fields are verified. As mentioned in the section 2.4, for (re)forecast fields, two different forecast fields from ECMWF (i.e., forecast based on WRF model and reforecast based on ECMWF model) are used for comparison. In particular, 24 h forecast fields of ERA5 using WRF model are evaluated, as shown in Figs. 4 and 7 in the revised manuscript. While 24 h reforecast fields of EARR (AdvHG) for 10-year period have been generated during production process for the whole period, producing 24 h forecast for ERA5 is not necessary. Hence, two-year period was originally chosen for the purpose of evaluation. Furthermore, we consider it more valuable to make our dataset open to the public at the earliest possible time, so that it could benefit more people using this dataset. For these reasons, two-year period was originally verified in this study.

Nevertheless, as the referee pointed out, more investigation for longer-term period would be necessary. Therefore, as the referee proposed, we have evaluated longer-term datasets of EARR and ERA5 that are able to be verified for the whole 10-year period and replaced the results of two-year period with those of ten-year period (Figs. 6 and 7 in the revised manuscript) (Please see Figs_rev2 and rev3 below). Because the aim of our study is to investigate EARR (AdvHG) performance with ERA5, it is worth evaluating reanalysis and (re)forecast fields of EARR and ERA5 for 10-year period, as referee pointed out. However, it seems unfeasible to compare EARR performance with various experiments (e.g., E3DVAR, ERA-Interim) for the whole period due to the high computational costs producing those datasets, especially ensemble-based one (i.e. E3DVAR). The updated results for the period of 2010-2019 (Figs_rev2 and rev3) are almost the same as the previous results with two-year period, except for the water vapor mixing ratio (Qvapor). Although Qvapor RMSEs of reanalysis and (re)forecast of EARR and ERA5 for ten-year period are greater than those for two-year period, both of EARR and ERA5 Qvapor RMSEs increase and the RMSE differences between EARR and ERA5 for ten-year period are

similar to those for two-year period. Thus, the longer-term evaluation reveals a large variability of atmospheric humidity and consequent predictability variability over East Asia for ten-year period of 2010-2019. We have revised the manuscript accordingly.

[Figure]

Figure_rev2. RMSEs of analysis of (a) zonal wind, (b) meridional wind, (c) temperature, and (d) Qvapor (water vapor mixing ratio) from ERA5 (black solid) and AdvHG (blue solid) and spreads of analysis (black dashed) and 6 h forecast (gray dashed) of AdvHG depending on pressure levels averaged over the ten-year period of 2010–2019.

[Figure]

Figure_rev3. Same as Fig_rev2 except for RMSE of 24 h forecast.

*4. The precipitation of AdvHG in winter (Juanuary) is the most accurate among other results, while E3DVAR (ERA5) performs better for weak (strong) thresholds in summer (July). The assimilation method tends to have positive effects on the simulated winter rainfall while brings limited improvement on summer rainfall. The possible underlying physics have not been well understood. Which physical process could be mostly possible affected leading to such positive effects?*

**Authors' response**: As the referee pointed out, AdvHG shows the most accurate performance of precipitation in winter. In contrast, it shows limited improvement for summer. For summer season, ETS differences between the different results are not significant and the values of ETS are approximately 2 times smaller than those for the winter season (Fig. 8 in the revised manuscript). Hence, the overall predictability of precipitation for winter season for all results is higher than that for summer season.

To elucidate which physical process is responsible for more accurate simulation of precipitation, non-convective and convective precipitation fields of AdvHG are investigated respectively. The precipitation fields from microphysics (AdvHG_NC) and cumulus (AdvHG_C) parameterization schemes in WRF model for January and July 2017 are shown in Figure_rev4, and two seasons have different features in precipitation. While the majority of precipitation in January 2017 is represented by the large-scale (non-convective) precipitation, the simulation of precipitation in July 2017 depends mostly on the convective precipitation. Because convective scheme is more likely to have larger sub-grid variability, the predictability of precipitation in winter where large-scale precipitation can represent most of the total precipitation fields tends to be higher than that in summer.

[Figure]

Figure_rev4. The spatial distribution of the monthly accumulated precipitation of AdvHG from microphysics (AdvHG_NC) and cumulus (AdvHG_C) schemes for January and July 2017.

Furthermore, although large-scale variables of ERA5 forecast based on WRF (wind, temperature, specific humidity) have the lowest RMSEs among other results both for winter and summer seasons, the differences of forecast RMSEs between ERA5 and AdvHG for

winter are much smaller than those for summer (Figs. 4 and 5 in the revised manuscript). The more accurate large-scale variables of AdvHG in winter could lead to higher AdvHG ETS in winter. Although AdvHG forecasts have less accurate large-scale variables than ERA5 forecasts, AdvHG ETS is higher than ERA ETS in winter implying that the precipitation could be a combination of small and large-scale processes in winter. In addition, it seems that producing analysis and forecast in the same system (WRF in AdvHG) could lead positive effects. Therefore, the consistency of modeling system for producing analysis and forecast could be fundamentally important.

*5. Figure 2, it is clear that E3DVAR has the lowest RMSE for up-level variables, so what are the advantages of AdvHG? Does ERA5 in Fig. 2 mean WRF results forcing by ERA5?*

**Authors' response**: As mentioned in the original manuscript L251-254 (in the revised manuscript L288-291), lower RMSEs of analysis fields do not guarantee the higher accuracy of analysis fields. Instead, RMSE of analysis indicates how much analysis fields are fitted to observations. Thus, not only those of analysis fields but also those of forecast fields need to be evaluated and compared to each other. As the referee pointed out in Fig. 2 in the original manuscript (Fig. 3 in the revised manuscript), the RMSEs of analysis fields of E3DVAR are lower than those of AdvHG. However, in Figs. 3 and 4 in the original manuscript (Figs 4 and 5 in the revised manuscript), RMSEs of 24 h and 36 h forecast fields of AdvHG are lower than those of E3DVAR. This is why AdvHG is used as a data assimilation method for East Asia Regional Reanalysis system in this study.

In addition, the ERA RMSEs in Fig. 2 in the original manuscript (Fig. 3 in the revised manuscript) are calculated based on ERA reanalysis fields downloaded from ECMWF. In contrast, ERA5 in Figs. 3, 4, and 6 in the original manuscript (Figs. 4, 5, and 7 in the revised manuscript) indicates forecast fields using WRF model, where ERA5 is used as the initial and lateral boundary conditions.

**Minor comments:**

*1. The meaning of variables in Eq. (3)-(6) should be clarified.*

**Authors' response**: Following the referee's suggestion, the explanation of variables in Eqs. (3)-(6) are added to the revised manuscript L141-143 as follows (underlined).

(L141-143) "$\mathbf{H}$ is an observation operator mapping the model state vector to observation space and $\mathbf{R}$ is the observation error covariance matrix. The matrices $\mathbf{P}^f$ and $\mathbf{B}$ indicate the ensemble-based and the static climatological BEC, respectively."

---

## Author Comment (AC2)

ESSD-2021-217
Response to RC2 (Referee 2)

The authors thank referee 2 for a thoughtful review of the manuscript. We agree with many of the referee's points and have made the necessary changes. The responses for the referee's specific comments are as follows.

*1. Title. Suggest the authors change it. If it is the development of EARR, the information of input observations data quality or evaluations of more variables/indexes during the whole 10 years 2010-2019 should be included in the figures, but not only 201701 and 201707 mainly in Fig 2-4, 7-13 and only 2017-2018 in Fig 4-5. If it is the development of AdvHG, the innovation contents from your own group should be included in 2.2.2 (page5-8), otherwise, they are all the approaches you could adopt, but not develop. The main contents of the paper are evaluation in 2017-2018, including the method, results and usage in EARR, so maybe it is more suitable to call the title like "Evaluation of EARR based on AdvHG", for your reference. If more figures of longer time series results could be replaced here, it is better. Anyway, the results are not enough, the period is short. The representativeness of the result is limited, comparing with the ERRA (2010-2019).*

**Authors' response**: We agree with the referee's point, so we have added or changed the contents according to the referee's suggestions in order to keep the original title for this manuscript.

Firstly, as the referee recommended, we have added the information on observations data quality to the revised manuscript as Table 2 (Table_rev1 below). Figure 1 is also modified to show spatial distributions of observations used in this study in the revised manuscript (Figure_rev1 below). Furthermore, the explanation of observation quality control procedure applied to this study is added to section 2.3 in the revised manuscript (underlined below).

Table_rev1. Summary of observations used in this study. The default observation error statistics provided in WRFDA system are used for assimilation in this study. The variables u, v, T, RH, Ps, and TPW denote zonal wind, meridional wind, temperature, relative humidity, surface pressure, and total precipitable water, respectively.

| Observations | Descriptions | Variables | Observation errors (depending on vertical levels) |
|---|---|---|---|
| SOUND | Upper-air observation from radiosonde | u, v | 1.1-3.3 m/s |
| | | T | 1 K |
| | | RH | 10-15% |
| PROFILER | Upper-air wind profile from wind profiler | u, v | 2.2-3.2 m/s |
| PILOT | Upper-air wind profile from pilot balloon or radiosonde | u, v | 2.2-3.2 m/s |

| AIREP | Upper-air wind and temperature from aircraft | u, v | 3.6 m/s |
| --- | --- | --- | --- |
| | | T | 1 K |
| Scatwind | Scatterometer oceanic surface winds | u, v | 2.5-3.8 m/s |
| SHIPS | Surface synoptic observation from ship | u, v | 1.1 m/s |
| | | T | 2 K |
| | | Ps | 1.6 hPa |
| | | RH | 10% |
| SYNOP | Surface synoptic observation from land station | u, v | 1.1 m/s |
| | | T | 2 K |
| | | Ps | 1 hPa |
| | | RH | 10% |
| BUOY | Surface synoptic observation from buoy | u, v | 1.4-1.6 m/s |
| | | T | 2 K |
| | | Ps | 0.9-1 hPa |
| | | RH | 10% |
| GPSPW | Precipitable water vapor from global positioning system (GPS) | TPW | 0.2 mm |
| METAR | Aviation routine weather report from automatic weather station (AWS) | u, v | 1.1 m/s |
| | | T | 2 K |
| | | Ps | 1 hPa |
| | | RH | 10% |
| AMV | Conventional atmospheric motion vector data from geostationary satellite | u, v | 2.5-4.5 m/s |

[Figure]

Figure_rev1. The East Asia Regional Reanalysis domain with different types of NCEP PrepBUFR observations available for assimilation at 00 UTC on 1st of January in 2017. The black dashed box denotes a verification area.

(L178-196) "The NCEP PrepBUFR [Prepared or QC'd data in BUFR (Binary Universal Form for the Representation of meteorological data) format] conventional observations (global upper air and surface weather observations, NCEP/NWS/NOAA/U.S.DOC 2008) are used every 6 h (00, 06, 12, and 18 UTC) for an assimilation by E3DVAR and AdvHG methods (Fig. 1). The PrepBUFR is the output of the final process for preparing the observations to be assimilated in the different NCEP analyses. For observations, rudimentary multi-platform quality control (QC) and more complex platform-specific QC were conducted (e.g., surface pressure, rawinsonde heights and temperature, wind profiler, aircraft wind and temperature) in NCEP (Keyser 2013). Furthermore, if the innovations (i.e., observation minus background) of some observations are greater than 5 times the observational error, then that observation is rejected during assimilation procedure in this study.

The assimilated observations are as follows: the surface observations (SYNOP, METAR, Ship, and Buoy), radiosonde observation (SOUND), upper-wind report (PILOT), wind profiler, aircraft, atmospheric motion vector (AMV) wind from a geostationary satellite (GEOAMV), scatterometer oceanic surface winds (Scatwind), and precipitable water vapor from global positioning system (GPSPW). The observation errors depending on each observation platform, variable, and vertical levels are assigned based on the default observation error statistics provided in WRFDA system (Table 2). All observations are spatially thinned by 20 km except for AMV thinned by 200 km as done by Warrick (2015), Cotton et al. (2016), and Shin (2016)."

Secondly, as referee proposed, we have evaluated longer-term datasets of EARR and ERA5 that are able to be verified for the whole 10-year period and replaced the results of two-year period with those of ten-year period (Figs. 6 and 7 in the revised manuscript) (Please see Figs_rev2 and 3 below). Because the aim of our study is to investigate EARR (AdvHG) performance with ERA5, it is worth evaluating reanalysis and (re)forecast fields of EARR and ERA5 for 10-year period, as referee pointed out. However, it seems unfeasible to compare EARR performance with various experiments (e.g., E3DVAR, ERA-Interim) for the whole period due to the high computational costs producing those datasets, especially ensemble-based one (i.e. E3DVAR). The updated results for the period of 2010-2019 (Figs_rev2 and 3) are almost the same as the previous results with two-year period, except for the water vapor mixing ratio (Qvapor). Although Qvapor RMSEs of reanalysis and (re)forecast of EARR and ERA5 for ten-year period are greater than those for two-year period, both of EARR and ERA5 Qvapor RMSEs increase and the RMSE differences between EARR and ERA5 for ten-year period are similar to those for two-year period. Thus, the longer-term evaluation reveals a large variability of atmospheric humidity and consequent predictability variability over East Asia for ten-year period of 2010-2019. We have revised the manuscript accordingly.

[Figure]

Figure_rev2. RMSEs of analysis of (a) zonal wind, (b) meridional wind, (c) temperature, and (d) Qvapor (water vapor mixing ratio) from ERA5 (black solid) and AdvHG (blue solid) and spreads of analysis (black dashed) and 6 h forecast (gray dashed) of AdvHG depending on pressure levels averaged over the ten-year period of 2010–2019.

[Figure]

Figure_rev3. Same as Fig_rev2 except for RMSE of 24 h forecast.

Lastly, we have added a new content from our group which is a schematic diagram of Advanced Hybrid Gain (AdvHG) method as Fig. 2 in the revised manuscript (Fig_rev4 below). And the section 2.2.2 for an explanation of AdvHG method is divided into two sections 2.2.2 and 2.2.3 to differentiate AdvHG method we newly developed in this study from the existing Hybrid Gain (HG) method.

[Figure]

$$X^a_{AdvHG} = \alpha X^{f(6h)}_{ERA5} + (1-\alpha)\overline{X}^a_{E3DVAR}$$

Figure_rev4. The schematic diagram of the advanced hybrid gain data assimilation method in the East Asia regional reanalysis system.

*2. Horizontal resolution, 12km. It is suggested to mention the raw description of model/DA like other reanalysis, for it is not the same resolution anywhere in the global. Add the information only once in 2.1, like in line 84 (540\*432 grid points), it is suggested.*

**Authors' response**: The model used in this study is the WRF model, which is a regional model based on a grid-point model not a spectral model. Thus, 12-km horizontal resolution for the WRF model is a reasonable way to express a horizontal grid spacing of a model. Meanwhile, for ERA-Interim and ERA5 models' resolution, as the referee suggested, we have added the information on spectral truncation (underlined) to the revised manuscript as follows.

(L205-206) "The horizontal resolutions of ERA-I and ERA5 are approximately 79 km (TL255) and 31 km (TL639), respectively."

*3. 1. introduction. The motivation is described well enough here, like a full story, while the scientific background introduction is not enough, not like an excellent scientific introduction in a paper.*

**Authors' response**: As the referee suggested, we have revised introduction to have the scientific background (underlined) in the revised manuscript.

(L50-63) "The long-term high-resolution datasets are essential to investigate the past extreme weather events which might be associated with mesoscale features such as heavy rainfall events with high spatial and temporal variability which coarser-resolution model cannot represent. The dynamical downscaling approaches can be a solution for generating high-resolution dataset, but they have some issues with insufficient spin-up (Kayaba et al. 2016). Moreover, Fukui et al. (2018) demonstrated that regional reanalysis over Japan assimilating only the conventional observations had the potential to reproduce precipitation fields better

than the dynamical downscaling approaches. Ashrit et al. (2020) also found that the high-resolution regional reanalysis over India showed substantial improvements of regional hydroclimatic features during summer monsoon for the period of 1979-1993 compared to the global reanalysis ERA-Interim (ERA-I, Dee et al. 2011) from ECMWF. Furthermore, He et al. (2019) revealed that the pilot regional reanalysis over the Tibetan Plateau was able to represent more accurate precipitation features as well as atmospheric humidity than the global reanalyses of ECMWF (i.e., ECMWF's fifth-generation reanalysis (ERA5, Hersbach et al. 2020) and ERA-I)."

*4. 2. system. Line 83-85, the sentence is not right, Fig 1 is the domain.*

**Authors' response**: As the referee pointed out, we have revised the sentence (underlined) in the revised manuscript as follows.

(L101-103) "In this study, the Advanced Research Weather Research and Forecasting (WRF, v3.7.1) model is used with 12-km horizontal resolution (540 x 432 grid points) and 50 vertical levels (up to 5 hPa) for East Asia domain shown in Fig. 1."

*5. Line 134-140, what is alpha in EQ 7?*

**Authors' response**: Alpha in Eq. (7) is a tunable parameter the same as in Eq. (2). To elucidate it, we have added a sentence (underlined) in the revised manuscript as follows.

(L156-158) "where $X_{ERA5}^{f(6h)}$ denotes the 6 h forecast of ERA5 reanalysis based on WRF model and $\overline{X}_{E3DVAR}^{a}$ denotes the analysis of E3DVAR (Fig. 2). In Eq. (7), α is a tunable parameter and is assigned to be 0.5 in this study."

*6. The authors make great effort in the DA approach, while what is the characteristics in East Asia of the EARR, comparing with other regional reanalysis, considering of the terrain, climate state like monsoon. In this scope, 50% is cover by the ocean, how is the ocean-land-atmospheric coupled here or just simply depends on all in WRF?*

**Authors' response**: There are a variety of regional reanalysis datasets particularly focusing on the impact of terrain like the Tibetan Plateau (He et al. 2019) and regional hydroclimatic features during monsoon over India (Ashrit et al. 2020). However, the main aim of this stage of this study is to develop a regional reanalysis over East Asia with newly proposed DA method and investigate the uncertainties and characteristics of general meteorological variables of the reanalysis such as temperature and precipitation with existing global reanalysis. Furthermore, we consider it more valuable to make our dataset open to the public at the earliest possible time, so that it could benefit more people using this dataset. As the referee pointed out, more investigation from different perspectives would be conducted in the future.

Ocean, land, and atmosphere are not coupled in WRF model, because WRF model is an atmospheric model producing atmospheric simulations. In this study, sea surface temperature (SST) obtained from ERA5 is used to be updated in WRF model and Unified Noah Land Surface Model (Tewari et al. 2004) is used as a land surface model.

*7. Line 160, it is wrong here to mention QuikSCAT which is 199907-200911, it is not in 2010-2019. For your reference: (1) Coriolis/WindSAT (20070813-20120804) from CFSR prepbufr, (2) Oceansat-2/OSCAT (20091215-20140220) KNMI reprocessed but not in CFSR, (3) MetOp-A/ASCAT (20070101-20140331 KNMI reprocessed , GTS data till present in GDAS), (4) MetOp-B/ASCAT (20140408-present in GDAS). You may not use reprocessed ASCAT wind, but it is used in ERA5.*

**Authors' response**: We appreciate the referee's correction. Even though QuikSCAT no longer collects ocean wind data, the convention of naming the scatterometer oceanic surface winds as a group of "QSCAT" still remains in WRFDA system, which made us mistaken. Because the scatterometer oceanic surface winds (Scatwind) are assimilated in EARR, we have rectified the error in the original manuscript by revising the sentence (underlined).

(L189-193) "The assimilated observations are as follows: the surface observations (SYNOP, METAR, Ship, and Buoy), radiosonde observation (SOUND), upper-wind report (PILOT), wind profiler, aircraft, atmospheric motion vector (AMV) wind from a geostationary satellite (GEOAMV), scatterometer oceanic surface winds (Scatwind), and precipitable water vapor from global positioning system (GPSPW)."

*8. 4. Result, it is suggest to shorten the results to 60%. The emphasis is how good is EARR but not ERA5. There are many sentences/paragraphs with the subject of ERA5 but not your reanalysis. Like line 235-256, 260-262, 290-291, 328-331, 410-412, 449-452, 520-521. And the difference between ERA5 and ERA-I could be shorten like line 265-270. Pay more attention in how good EARR but not how is ERA5 like line 357-363. The order of the results from different reanalysis is also important. Line 312-314 is good in expression.*

**Authors' response**: Following the referee's suggestion, we have shortened the results and revised the manuscript to pay more attention to EARR rather than ERA5.

*9. Line 324, except for strong thresholds, how is strong? >4 mm/6h? how is week?*

**Authors' response**: As the referee pointed out, we have revised sentences (underlined) in the revised manuscript as follows. For a more objective and specific description, the adjectives for threshold such as 'strong' and 'weak' are replaced by 'high' and 'low' and specific threshold values are presented in the revised manuscript.

(L353-356) "For January 2017 (Fig. 8a), ETS of ERA5 (i.e., WRF-based ERA5) is higher than that of ERA5_fromECMWF for all thresholds, whereas ETS of ERA-I (i.e., WRF-based

ERA-I) is lower than that of ERA-I_fromECMWF except for high thresholds (8 and 16 mm (6 h)$^{-1}$)."

(L360-362) "Regarding FBI in winter (Fig. 8b), for 4, 8, and 16 mm (6 h)$^{-1}$ thresholds, all the results show the FBI smaller than 1, implying the underestimation of frequency of precipitation for high-threshold events."

(L374-376) "With respect to FBI in July 2017, the WRF-based results show the FBIs greater than 1, whereas reforecast from ECMWF show the FBIs greater than 1 for 0.5, 1, and 4 mm (6 h)$^{-1}$ thresholds and smaller than 1 for higher thresholds (8 and 16 mm (6 h)$^{-1}$) (Fig. 8d)."

(L417-420) "At 0.5, 1, and 4 mm (6 h)$^{-1}$ thresholds, E3DVAR BSS is the greatest, which is similar to ETS. At 8 and 16 mm (6 h)$^{-1}$ thresholds, ERA5 ETS is the highest, followed by AdvHG and E3DVAR, whereas overall E3DVAR BSS is the highest, followed by AdvHG and ERA5."

(L440-442) "During July in 2017, ERA5 and ERA-I simulate heavier precipitation than AdvHG (not shown), which is consistent with larger FBI of ERA5 and ERA-I at higher thresholds."

(L456-460) "Moreover, although all the results similarly represent overall features of precipitation in January (Fig. 13), ERA5_fromECMWF (Fig. 13g) simulates the overestimated precipitation over South China, which is consistent with the results in the previous section as well as its larger FBI at lower thresholds (0.5 and 1 mm (6 h)$^{-1}$) shown in Fig. 8b."

(L468-470) "This is consistent with the result in Fig. 8d, in which FBIs from WRF-based results are generally greater than 1 for higher thresholds (8 and 16 mm (6 h)$^{-1}$), whereas those from ECMWF are smaller than 1."

(L538-542) "In addition, the ETS differences between the results are not distinctive in July. For higher thresholds (8 and 16 mm (6 h)$^{-1}$) in July, AdvHG ETS is greater than E3DVAR ETS and smaller than ERA5 ETS, whereas E3DVAR ETS is the greatest followed by ERA5 and AdvHG for lower thresholds (0.5 and 1 mm (6 h)$^{-1}$)."

*10. Fig 7, Line 346-348, results of AdvHG in Jan is better than in Jul, FBI closer to 1. Different FBI results in Jul are larger than 1 (over-forecast) more than in Jan, more difficult to improve for summer than winter. Index ETS and FBI are more difficult to handle and analysis than POD and FAR which is better when it is larger and smaller, separately.*

**Authors' response**: POD and FAR seem to be straightforward to deal with. However, as mentioned in section 3.2, POD can be artificially improved by systematically over-forecasting the events (Wilson 2010), so FAR should be used with POD. Moreover, ETS is a more balanced score than POD and FAR, because it is sensitive to both false alarms and

misses (Wilson 2010). To elucidate this, we have added the explanation (underlined) about ETS in the revised manuscript.

(L223-225) "The ETS range is from -1/3 to 1 and the value 1 for ETS is a perfect score. ETS is a more balanced score than Probability of Detection (POD) and False Alarm Ratio (FAR), because it is sensitive to both false alarms and misses (Wilson 2010)."

*11. Line 354, (Figs. 8a and b), is it right? 8b is FAR.*

**Authors' response**: As the referee pointed out, we have revised the sentence (underlined) in the revised manuscript as follows.

(L383-384) "For January in 2017, AdvHG POD is the greatest among the WRF-based results, followed by E3DVAR, ERA5, and ERA-I (Fig. 9a)."

*12. 6. Summary, it is not concise in this paragraph.*

**Authors' response**: As the referee pointed out, we have revised the summary in the revised manuscript to make it concise.

*13. Reference, line 608-614, it is repeated, please delete. Line 621-622 seems with larger font size.*

**Authors' response**: As the referee suggested, we have deleted the repeated reference and reduced the font size of the reference the referee pointed out in the revised manuscript.

*14. Fig 2, is it the result of all EARR domain? Jan and Jul shown in Fig 2-6 while YYYYMM or YYYYMMDD shown in Fig 7-11, it is suggested to use unified expression. Fig 6, Temp revised to T like Fig 2.*

**Authors' response**: Figure 2 in the original manuscript (Fig. 3 in the revised manuscript) is the result of verification domain (dashed box in Fig. 1), not all EARR domain. To make it clearer, we have added the explanation about verification domain in the revised manuscript as follows (underlined).

(L272-276) "The analysis and forecast RMSEs of E3DVAR, AdvHG, the WRF-based ERA-I, and WRF-based ERA5 are calculated for zonal wind, meridional wind, temperature, and Qvapor (water vapor mixing ratio in WRF) variables against sonde observations at 00 and 12 UTC in verification domain (dashed box in Fig. 1) for January and July in 2017 and averaged over each month (Figs. 3, 4, and 5)."

Furthermore, as the referee suggested, we have modified Figs 3-7 to use unified expression (e.g., YYYYMM) and have changed 'Temp' to 'T' in Fig. 7 in the revised manuscript.

**References**

He, J., F. Zhang, X. Chen, X. Bao, D. Chen, H. M. Kim, H.-W. Lai, L. R. Leung, X. Ma, Z. Meng, T. Ou, Z. Xiao, E.-G. Yang, and K. Yang, 2019: Development and evaluation of an ensemble-based data assimilation system for regional reanalysis over the Tibetan Plateau and surrounging regions, *Journal of Advances in Modeling Earth Systems*, **11**(8), 2503-2522.

Kayaba, N., T. Yamada, S. Hayashi, K. Onogi, S. Kobayashi, K. Yoshimoto, K. Kamiguchi, and K. Yamashita, 2016: Dynamical regional downscaling using the JRA-55 reanalysis (DSJRA-55). *Sola*, **12**, pp.1-5.

Keyser, D., 2013: An Overview of Observational Data Processing at NCEP (with information on BUFR Format including "PrepBUFR" files), *GSI tutorial*, August 6, 2013.

Tewari, M., F. Chen, W. Wang, J. Dudhia, M. A. LeMone, K. Mitchell, M. Ek, G. Gayno, J. Wegiel, and R. H. Cuenca, 2004: Implementation and verification of the unified NOAH land surface model in the WRF model. *20th conference on weather analysis and forecasting/16th conference on numerical weather prediction* (Vol. 1115). Seattle, WA: American Meteorological Society.

Wilson, L., 2010: Verification of severe weather forecasts in support of the "SWFDP Southern Africa" project. Rep. for the WMO, 21 pp., www.wmo.int/pages/prog/www/BAS/documents/Doc-7-Verification.doc.

---

## Author Comment (AC3)

ESSD-2021-217
Response to CC1 (Yan Liu)

The authors thank Yan Liu for a thoughtful review of the manuscript. The response for the community comments is as follows.

**Comments:**

*The authors have done a lot of works to develop a regional reanalysis system, which are worth publishing. However, the reanalysis is to use more and more reliable archived historical observations than those obtained from real-time numerical forecast analysis, as well as newer and better numerical forecast models and assimilation systems to produce higher-quality analysis. The development of reanalysis system should include two parts, numerical weather prediction system and observations. The authors should present more advantages of EARR compared to other regional reanalysis system. Otherwise, the value of the new regional reanalysis system is less. For example, ERA5 reanalysis has rich satellite data with 25Km resolution, EARR only assimilates conventional observations and Quick scant wind but half domain of ERRA is ocean. ERRA uses ERA5 as initial value and LBC, the forecast has much information of ERA5. It is better to provide some experiments to support the reason why use ERA5 forecast field is better than deterministic analysis in page 7.*

    **Authors' response**: Specific responses for Yan Liu's comments are as follows.

*A. However, the reanalysis is to use more and more reliable archived historical observations than those obtained from real-time numerical forecast analysis, as well as newer and better numerical forecast models and assimilation systems to produce higher-quality analysis. The development of reanalysis system should include two parts, numerical weather prediction system and observations.*

    **Authors' response**: As Yan Liu mentioned, it is essential to present the numerical weather prediction system including a data assimilation method and observations used to develop a reanalysis system. To elucidate the data assimilation system used in East Asia Regional Reanalysis system (EARR), we have added the schematic diagram of the advanced hybrid gain data assimilation method as Figure_rev1 below (Fig. 2 in the revised manuscript).

[Figure]

$$X_{AdvHG}^a = \alpha X_{ERA5}^{f(6h)} + (1-\alpha)\overline{X}_{E3DVAR}^a$$

Figure_rev1. The schematic diagram of the advanced hybrid gain data assimilation method in the East Asia regional reanalysis system.

To provide more information on observations assimilated in EARR, we have added details of observations data quality to the revised manuscript as Table 2 (Table_rev1 below). Figure 1 is also modified to show spatial distributions of observations used in this study in the revised manuscript (Figure_rev2 below). Furthermore, the explanation of observation quality control procedure applied to this study is added to section 2.3 in the revised manuscript (underlined below).

Table_rev1. Summary of observations used in this study. The default observation error statistics provided in WRFDA system are used for assimilation in this study. The variables u, v, T, RH, Ps, and TPW denote zonal wind, meridional wind, temperature, relative humidity, surface pressure, and total precipitable water, respectively.

| Observations | Descriptions | Variables | Observation errors (depending on vertical levels) |
|---|---|---|---|
| SOUND | Upper-air observation from radiosonde | u, v | 1.1-3.3 m/s |
| | | T | 1 K |
| | | RH | 10-15% |
| PROFILER | Upper-air wind profile from wind profiler | u, v | 2.2-3.2 m/s |
| PILOT | Upper-air wind profile from pilot balloon or radiosonde | u, v | 2.2-3.2 m/s |
| AIREP | Upper-air wind and temperature from aircraft | u, v | 3.6 m/s |
| | | T | 1 K |
| Scatwind | Scatterometer oceanic surface winds | u, v | 2.5-3.8 m/s |
| SHIPS | Surface synoptic observation from ship | u, v | 1.1 m/s |
| | | T | 2 K |
| | | Ps | 1.6 hPa |
| | | RH | 10% |
| SYNOP | Surface synoptic observation from land station | u, v | 1.1 m/s |
| | | T | 2 K |
| | | Ps | 1 hPa |
| | | RH | 10% |

| | | | |
|---|---|---|---|
| BUOY | Surface synoptic observation from buoy | u, v | 1.4-1.6 m/s |
| | | T | 2 K |
| | | Ps | 0.9-1 hPa |
| | | RH | 10% |
| GPSPW | Precipitable water vapor from global positioning system (GPS) | TPW | 0.2 mm |
| METAR | Aviation routine weather report from automatic weather station (AWS) | u, v | 1.1 m/s |
| | | T | 2 K |
| | | Ps | 1 hPa |
| | | RH | 10% |
| AMV | Conventional atmospheric motion vector data from geostationary satellite | u, v | 2.5-4.5 m/s |
| | | | |

[Figure]

Figure_rev2. The East Asia Regional Reanalysis domain with different types of NCEP PrepBUFR observations available for assimilation at 00 UTC on 1st of January in 2017. The black dashed box denotes a verification area.

(L178-196) "The NCEP PrepBUFR [Prepared or QC'd data in BUFR (Binary Universal Form for the Representation of meteorological data) format] conventional observations (global upper air and surface weather observations, NCEP/NWS/NOAA/U.S.DOC 2008) are used every 6 h (00, 06, 12, and 18 UTC) for an assimilation by E3DVAR and AdvHG methods (Fig. 1). The PrepBUFR is the output of the final process for preparing the observations to be assimilated in the different NCEP analyses. For observations, rudimentary multi-platform quality control (QC) and more complex platform-specific QC

were conducted (e.g., surface pressure, rawinsonde heights and temperature, wind profiler, aircraft wind and temperature) in NCEP (Keyser 2013). Furthermore, if the innovations (i.e., observation minus background) of some observations are greater than 5 times the observational error, then that observation is rejected during assimilation procedure in this study.

The assimilated observations are as follows: the surface observations (SYNOP, METAR, Ship, and Buoy), radiosonde observation (SOUND), upper-wind report (PILOT), wind profiler, aircraft, atmospheric motion vector (AMV) wind from a geostationary satellite (GEOAMV), scatterometer oceanic surface winds (Scatwind), and precipitable water vapor from global positioning system (GPSPW). The observation errors depending on each observation platform, variable, and vertical levels are assigned based on the default observation error statistics provided in WRFDA system (Table 2). All observations are spatially thinned by 20 km except for AMV thinned by 200 km as done by Warrick (2015), Cotton et al. (2016), and Shin (2016)."

In addition, since we liked to get information from more observation data than those obtained from real-time numerical forecast analysis, we combined the E3DVAR analysis and ERA5 forecast as in Eq. (7) in the manuscript. This part is explained in the response to the next comment in more detail.

*B. The authors should present more advantages of EARR compared to other regional reanalysis system. Otherwise, the value of the new regional reanalysis system is less. For example, ERA5 reanalysis has rich satellite data with 25Km resolution, EARR only assimilates conventional observations and Quick scant wind but half domain of ERRA is ocean. EARR uses ERA5 as initial value and LBC, the forecast has much information of ERA5.*

**Authors' response**: The EARR uses much more information than the E3DVAR using WRF, since we combined two information from E3DVAR and ERA5 to produce the EARR. EARR uses ERA5 as LBC, but combines E3DVAR analysis and ERA5 6 h forecast to get initial condition. In order to take advantage of more observations and advanced data assimilation method used for ERA5, a new advanced hybrid gain (AdvHG) data assimilation method, which combines E3DVAR and ERA5 based on WRF model, is newly proposed and investigated in this study.

As we mentioned in the manuscript, this is a very efficient approach because of the cost savings as well as the use of the high-quality latest reanalysis from ECMWF assimilating all currently available observations with the state-of-the-art and advanced technology. As a result, the precipitation of EARR is shown to be more accurate than that of ERA5 for both summer and winter seasons over East Asia. In a regional sense, a higher resolution regional-based reanalysis considering regional weather and climate characteristics is more and more required, and the method presented in this study shows the possibility of integrating various data, observations, and methodologies to suit regional needs.

*C. It is better to provide some experiments to support the reason why use ERA5 forecast field is better than deterministic analysis in page 7.*

**Authors' response**: In this study, 6 h forecast of ERA5 based on WRF model is used instead of ERA5 reanalysis fields to maintain the consistency between different modeling systems as well as different resolutions. ERA5 reanalysis fields are generated based on the Integrated Forecasting System (IFS) of ECMWF with around 30 km horizontal resolution. If ERA5 reanalysis is directly used to combine with E3DVAR analysis based on WRF model whose horizontal resolution is 12 km, there could be some imbalance occurred in meteorological fields resulting from two different modeling systems. Thus, to reduce the imbalance and ensure the stability and consistency during analysis process, 6 h forecast of ERA5 is used instead of ERA5 deterministic reanalysis in Advanced Hybrid Gain method.

---

## Author Response (AR2)

ESSD-2021-217
Response to RC1 (Referee 1)

The authors thank referee 1 for a thoughtful review of the manuscript. We agree with many of the referee's points and have made the necessary changes. The responses for the referee's specific comments are as follows.

**Overall comments:**

*The authors have made some substantial improvements to the manuscript, and have addressed most of my concerns. While there are still several issues that should be addressed before publication. Thus, I recommend major revisions.*

**Major comments:**

*1. The authors have provided more information on the experimental design, however, the description of the experiments used in this study is still limited. Does the WRF-ERA5 experiment generate a 36-hour simulation for each run, driven by ERA5 as the initial and lateral boundary conditions? Does it run once per day? If so, the reinitialization method is used. More details should be given on the experiments used in the study.*

> **Authors' response**: Following the referee's suggestion, we have added more information on the experiments to Table 3 in the revised manuscript to clear up the ambiguity.

> In this study, the WRF-based ERA5 experiment produces 36 hr forecasts from the initial condition (i.e., ERA5) every 6 hr (i.e., 00, 06, 12, and 18 UTC), which implies that there are 4 runs per day, and ERA5 reanalysis fields are used as the initial and lateral boundary conditions. This can be regarded as the re-initialization method. However, the term "re-initialization" is generally used in regional climate simulation studies and re-initialization frequency varies from a few days to a month. In contrast, in this study, the WRF-based ERA5 and ERA-I experiments are performed to be compared with data assimilation experiments (i.e., E3DVAR and AdvHG) and 36 hr forecast fields are generated every 6 hr for the experiments, which implies that forecast length (36 hr) is much shorter and re-initialization frequency (6 hr) is more frequent than general regional climate model (RCM) simulations where weekly or monthly re-initialization is applied for longer integration time. Furthermore, the comparison between experiments in this research is conducted to investigate the effect of assimilation and short-term predictability rather than climate simulations. For these reasons, the term "re-initialization" may not seem to be apposite to the description for experiments generated in this study.

*2. I don't agree with the author's reply to comment 5. Figures 4 and 5 in the revised manuscript show that WRF-ERA5 presents the smallest RMSEs for both the 24h and 36h forecast fields. I am wondering what are the advantages of high-resolution regional reanalysis.*

**Authors' response**: Although WRF-ERA5 shows the smallest RMSEs for 24 and 36 hr forecast fields, precipitation fields of WRF-ERA5 are less accurate than those of East Asia Regional Reanalysis (EARR). It may result from not only inconsistency between ECMWF modeling system to produce ERA5 and WRF model to produce forecast fields of ERA5 in this study, but also lack of smaller-scale features in an initial condition due to a relatively coarse resolution of ERA5. On the contrary, EARR uses the advanced hybrid gain data assimilation method (AdvHG) to add value to ERA5 global reanalysis. Combining the global reanalysis data (i.e., ERA5) characterized by the high quality of large-scale features with detailed smaller-scale features in the higher resolution represented by ensemble-based assimilation method (i.e., E3DVAR) as well as a community numerical weather prediction model (i.e., WRF model) is a key factor of EARR to be able to produce high-resolution initial conditions represented with regional features, which could contribute to reduction of forecast errors, especially for precipitation. Therefore, EARR has its own advantage of representing regional features of precipitation better than relatively coarse-resolution global reanalysis. To highlight the advantages of EARR, we have added them to section 6 in the revised manuscript as follows (underlined).

(L569-576) "Combining the global reanalysis data (i.e., ERA5) characterized by the high quality of large-scale features with detailed smaller-scale features in the higher resolution represented by ensemble-based assimilation method (i.e., E3DVAR) as well as a community numerical weather prediction model (i.e., WRF model) is a key factor of EARR to be able to produce high-resolution initial conditions represented with regional features, which could contribute to reduction of forecast errors, especially for precipitation. Therefore, EARR has its own advantage of representing regional features of precipitation better than relatively coarse-resolution global reanalysis."

ESSD-2021-217
Response to RC2 (Referee 2)

The authors thank referee 2 for a thoughtful review of the manuscript. We agree with many of the referee's points and have made the necessary changes. The responses for the referee's specific comments are as follows.

**Overall comments:**

*The authors revised the paper well enough, especially the results of 2010-2019. Readers will benefit the careful revise. Some tiny problems are below, for your reference.*

**Minor comments:**

*1. Line 191 and Table 1. The conventional data includes AMVs, it should be AMVs from geostationary satellites and polar winds from polar winds. Suggest not to use "GEOAMV" in Line 191, and "geostationary" in Table1. Please check the actual data input again.*

> **Authors' response**: As the referee suggested, we have checked the actual AMV input data from NCEP Prepbufr observations and have revised Table 2 and Line 191 in the revised manuscript. Because most of the EARR domain is lower than 60 degrees north and polar orbiting satellites generally cover high latitude areas beyond 60 degrees north, in most cases, AMVs from geostationary satellites (Himawari-8 and Meteosat-8) account for most of AMVs in the EARR domain as shown in Figure_rev1. Each dot color in Figure_rev1 represents different report types of AMV winds from NCEP PrepBUFR observations available for assimilation at 00 UTC on 1st of January in 2018. The reddish colored dots for 242, 250, and 252 indicate AMV winds from Himawari-8 satellite and the bluish colored dots for 243, 253, and 254 indicate those from Meteosat-8 satellite. Although polar AMVs are rarely included in the EARR domain, it is more accurate to use the term AMV instead of GEOAMV. Therefore, we have revised Table 2 and Line 191 in the revised manuscript accordingly.

[Figure]

Figure_rev1. Each dot color represents different report types of AMV winds from NCEP PrepBUFR observations available for assimilation at 00 UTC on 1st of January in 2018. The reddish colored dots for 242, 250, and 252 indicate AMV winds from Himawari-8 satellite and the bluish colored dots for 243, 253, and 254 indicate those from Meteosat-8 satellite.

*2. Delete "2017010100" on the upper right of Figure 1. Enough information in the title of the figure 1. Please avoid the misunderstanding to the readers. The scope is applied for 10 years. Please revise the title, the following is for your reference.*

*Figure 1. The East Asia Regional Reanalysis domain. The black dashed box denotes a verification area. Different types of NCEP PrepBUFR observations are available for assimilation at 00 UTC on 1st of January in 2017.*

**Authors' response**: As the referee suggested, we have revised Fig. 1 and its title in the revised manuscript.